# Elucidating the rate-limiting step of CO$_2$ electroreduction on metal phthalocyanines

Zhuanghe Ren[1], Kaige Shi [1], Zhen Meng [2], Thomas Egan[2], Talat S. Rahman [1,3] & Xiaofeng Feng [1,2,3,4] ✉

Immobilized molecular catalysts, especially metal phthalocyanines, have garnered substantial interest for the electrochemical CO$_2$ reduction reaction (CO$_2$RR) due to their well-defined active sites and promising performance. Yet, the reaction mechanism, particularly the rate-limiting step, remains debated. Here, using electrochemical analysis and kinetic isotope effect measurements, we identify the rate-limiting step of CO$_2$RR to CO on immobilized metal phthalocyanines, with Au as a reference. Notably, cobalt phthalocyanine (CoPc) exhibits dispersion-dependent kinetics: protonation of adsorbed *CO$_2$ is rate-limiting on molecularly dispersed CoPc supported on carbon nanotubes (CoPc/CNTs), whereas CO$_2$ adsorption becomes rate-limiting on aggregated CoPc due to a weakened interfacial electric field at the Co active sites. This mechanistic distinction further elucidates the role of electrolyte anions: HCO$_3^-$, largely a spectator on Au, promotes CO$_2$RR on CoPc/CNTs by serving as a proton donor in the rate-limiting protonation step. These findings provide mechanistic insights into CO$_2$RR on metal phthalocyanines and guide the rational design of molecular electrocatalysts.

Electrochemical CO$_2$ reduction reaction (CO$_2$RR) powered by renewable electricity offers a promising pathway toward carbon neutrality by transforming abundant carbon emissions into value-added chemicals[1–3]. To achieve the techno-economic viability of this process, it is essential to develop highly active and selective catalysts[4]. Metallic materials have been widely employed to catalyze CO$_2$RR, resulting in a range of products including CO[5–8], formate[9,10], and multi-carbon hydrocarbons and oxygenates such as ethylene, ethanol, and propanol[11–14]. Although substantial progress has been made in improving the CO$_2$RR performance of metal catalysts, the inherent multiplicity of active sites hinders a clear mechanistic understanding and rational catalyst design. Another challenge for metal catalysts is to achieve high CO$_2$RR selectivity due to the competing hydrogen evolution reaction (HER)[15,16], especially in acidic media. In contrast, molecular catalysts with well-defined metal centers have demonstrated notable activity for CO$_2$RR and are relatively inert toward HER, providing a promising alternative to metal catalysts[17].

Molecular catalysts with a variety of transition metal centers and macrocyclic structures have been investigated for CO$_2$RR[18–21]. They are typically immobilized on carbon substrates and exhibit promising catalytic performance for the CO$_2$RR to CO. For instance, Faradaic efficiencies for CO production exceeding 90% at relatively low over-potentials have been reported for the CO$_2$RR on cobalt phthalocyanines (CoPc)[22–24], cobalt porphyrins[25,26], cobalt quaterpyridine complexes[27], nickel phthalocyanines (NiPc)[28], and functionalized iron porphyrins[29,30]. Among these molecules, metal phthalocyanines have been the most extensively studied due to their facile availability, high CO$_2$RR activity, and capability of producing more reduced products such as methanol[31–35]. Despite the advances in catalytic performance, the reaction mechanism, particularly the rate-limiting step, remains elusive for CO$_2$RR on metal phthalocyanines[36–39].

For CO$_2$RR to CO on metal catalysts, CO$_2$ adsorption is generally considered to be the rate-limiting step, as schematically illustrated in Fig. 1a. However, on metal phthalocyanines such as CoPc, both CO$_2$

[1]Department of Physics, University of Central Florida, Orlando, FL, USA. [2]Department of Chemistry, University of Central Florida, Orlando, FL, USA. [3]Renewable Energy and Chemical Transformations (REACT) Cluster, University of Central Florida, Orlando, FL, USA. [4]Department of Materials Science and Engineering, University of Central Florida, Orlando, FL, USA. ✉e-mail: Xiaofeng.Feng@ucf.edu

adsorption and subsequent protonation of adsorbed *$CO_2$ have been proposed as rate-limiting steps (Fig. 1b). McCrory and co-workers evaluated the kinetic isotope effect (KIE) for the $CO_2RR$ on CoPc immobilized on a graphite disk electrode[37]. They found that the $CO_2RR$ activity remained nearly identical in aprotic and deuterated phosphate electrolytes, indicative of a rate-limiting step without proton involvement, and consequently, identified the $CO_2$ adsorption step as rate-determining. A similar result was reported for $CO_2RR$ on CoPc supported on carbon nanotubes (CNTs)[32]. In contrast, Chan and co-workers suggested that $CO_2RR$ on CNT-supported CoPc is limited by the protonation of adsorbed *$CO_2$ through a combination of computations and pH-dependent measurements[38]. This conclusion was supported by the work of Feng et al., in which the KIE studies showed a significantly higher $CO_2RR$ activity in $H_2SO_4$ compared to that in $D_2SO_4$, indicating that the protonation of *$CO_2$ is the rate-limiting step[39]. Given these controversial results, it is crucial to elucidate the rate-limiting step of $CO_2RR$ on metal phthalocyanines.

In this work, we investigate and elucidate the rate-limiting step of the $CO_2RR$ to CO on metal phthalocyanines, with Au catalyst as a reference. By combining electrochemical measurements and KIE analysis, we show that the $CO_2RR$ on molecularly dispersed CoPc and NiPc supported on CNTs (CoPc/CNTs and NiPc/CNTs) is limited by the protonation of adsorbed *$CO_2$. Using CoPc as a model system, we further demonstrate that the rate-limiting step of $CO_2RR$ depends on the catalyst's dispersion state, shifting to $CO_2$ adsorption on aggregated CoPc due to a weakened interfacial electric field at the Co active sites. Considering the rate-limiting protonation step on CoPc/CNTs, we further examine the often-overlooked role of electrolyte anions in $CO_2RR$ and reveal a more pronounced effect of $HCO_3^-$ relative to $Na^+$, in contrast to observations on typical metal catalysts. The promotional effect of $HCO_3^-$ is attributed to its role as a proton donor in the protonation of *$CO_2$. Our work clarifies the reaction mechanism and rate-limiting step of the $CO_2RR$ on metal phthalocyanines, thereby laying the groundwork for the rational design of efficient molecular electrocatalysts.

## Results

### Characterization of CoPc/CNTs and NiPc/CNTs catalysts

Molecular catalysts are typically integrated into electrodes by pre-immobilization or direct deposition onto carbon nanomaterials to achieve molecular dispersion[18]. When anchored onto CNT supports, the molecular catalysts can exhibit high activity and selectivity for $CO_2RR$, which is attributed to the increased exposure of active sites and enhanced electron transfer[18]. Therefore, we prepared CoPc/CNTs and NiPc/CNTs samples using a well-established anchoring strategy (see Methods for details)[31]. To contextualize our findings within the broader $CO_2RR$ field, we selected Au as a benchmark catalyst. Therefore, Au nanoparticles supported on carbon black (Au/C) were also prepared to serve as a reference due to its well-understood catalytic

mechanism for $CO_2RR$. The three samples were characterized for their morphology, structure, and chemical state. As shown in Fig. 2a, d, scanning transmission electron microscopy (STEM) images and corresponding energy-dispersive X-ray spectroscopy (EDS) mapping indicate a homogeneous dispersion of CoPc and NiPc on CNTs in the CoPc/CNTs and NiPc/CNTs samples, respectively. High-resolution X-ray photoelectron spectra (XPS) of Co 2$p$ and N 1$s$ regions (Fig. 2b, c) as well as Ni 2$p$ and N 1$s$ regions (Fig. 2e, f) confirm the presence of CoPc and NiPc molecules in the samples. Moreover, both scanning electron microscopy (SEM) and transmission electron microscopy (TEM) images indicate that the morphology of CoPc/CNTs and NiPc/CNTs resembles that of the pristine CNTs, showing no visible molecular aggregates (Supplementary Figs. 1 and 2). Their X-ray diffraction (XRD) patterns further support the absence of crystalline aggregates of CoPc and NiPc, as their characteristic diffraction peaks were not observed (Supplementary Figs. 1 and 2). Collectively, these results confirm a molecular dispersion state of CoPc and NiPc on CNTs. For the Au/C nanopowder, TEM, XRD, and XPS characterizations indicate an average particle size of around 5 nm and a metallic state of the Au nanoparticles, as shown in Supplementary Fig. 3. Each of the three samples was deposited onto an AvCarb GDS2230 carbon substrate to form the corresponding electrode.

### Identification of the rate-limiting step

To identify the rate-limiting step of $CO_2RR$, we employ the KIE method, as it can determine proton involvement in the key step, thereby distinguishing between the steps of $CO_2$ adsorption and protonation of adsorbed *$CO_2$. The magnitude of KIE is defined by $j_H/j_D$, where $j_H$ is the partial current density for a specific product in a protic electrolyte and $j_D$ is that in a deuterated electrolyte[37]. If the rate-limiting step is a protonation process, a KIE value >1 will be observed due to the sluggish dissociation of deuterated water and deuteron transfer process. In contrast, a rate-limiting step without proton involvement should result in a KIE value of around 1. To evaluate the KIE, $CO_2RR$ bulk electrolysis was performed in a two-compartment electrochemical cell (H-cell, Supplementary Fig. 4) with 1 M $NaHCO_3$ in $H_2O$ and 1 M $NaDCO_3$ in $D_2O$ electrolytes, respectively. All potentials were 100% $iR$-compensated using the current-interrupt method (unless otherwise specified) and are reported on the reversible hydrogen electrode (RHE) scale throughout this work. Gas-phase products, including CO and $H_2$, were quantified using gas chromatography (GC), and liquid-phase products were analyzed by $^1H$ nuclear magnetic resonance (NMR) spectroscopy.

As shown in Fig. 3a, in both $NaHCO_3/H_2O$ and $NaDCO_3/D_2O$ electrolytes, the partial current density for CO production on the Au/C electrode gradually increased from 3.8 to 10.9 mA cm$^{-2}$ as the potential shifted from −0.45 to −0.60 V versus RHE, resulting in KIE values of around 1 (Fig. 3b). Meanwhile, the CO Faradaic efficiency was 86–89% in the $NaHCO_3/H_2O$ electrolyte, but increased to 95–97% in $NaDCO_3/D_2O$, as presented in Fig. 3c. This is because the $D_2$ partial current

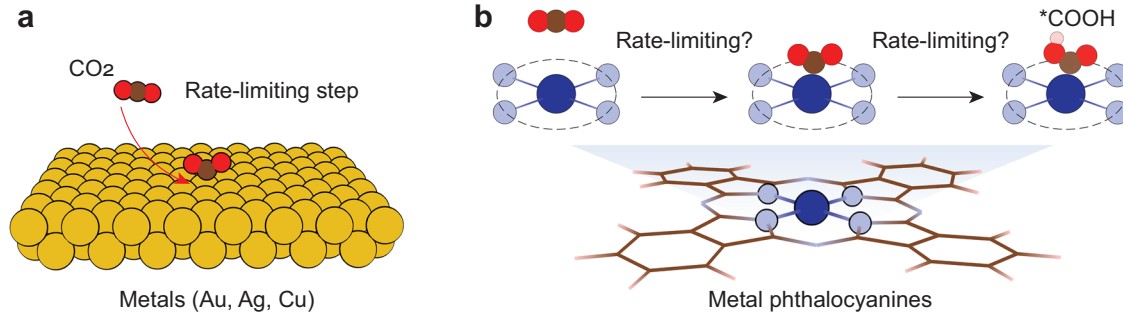

**Fig. 1 | Proposed rate-limiting steps of the $CO_2RR$ on different catalysts. a** $CO_2$ adsorption is identified as the rate-limiting step for the $CO_2RR$ to CO on typical metal catalysts. **b** For metal phthalocyanines, both $CO_2$ adsorption and the subsequent protonation of adsorbed *$CO_2$ have been proposed as rate-limiting steps.

densities in NaDCO$_3$/D$_2$O were significantly lower than H$_2$ in NaHCO$_3$/H$_2$O, as shown in Supplementary Fig. 5, leading to a higher CO Faradaic efficiency in NaDCO$_3$/D$_2$O. This confirms the reliability of the KIE method for determining proton involvement in the rate-limiting step. The result indicates that the CO$_2$RR on Au nanoparticles is limited by the CO$_2$ adsorption step without proton involvement, in consistent with previous studies[38,40]. In contrast, the CoPc/CNTs electrode exhibited consistently higher CO partial current densities in NaHCO$_3$/H$_2$O than in NaDCO$_3$/D$_2$O over the potential range of −0.55 to −0.70 V versus RHE (Fig. 3d), leading to KIE values of 1.4–1.5 (Fig. 3e). The CO Faradaic efficiencies on CoPc/CNTs in NaDCO$_3$/D$_2$O were higher than those in NaHCO$_3$/H$_2$O (Fig. 3f), which is similar to the case of Au catalyst, due to the lower D$_2$ production rates in NaDCO$_3$/D$_2$O. These results suggest that the protonation of adsorbed *CO$_2$ is the rate-limiting step of CO$_2$RR on CoPc/CNTs. Similarly, on the NiPc/CNTs electrode, CO production rate was generally higher in NaHCO$_3$/H$_2$O than in NaDCO$_3$/D$_2$O, yielding KIE values of 1.3–1.5, as shown in Fig. 3g, h. Meanwhile, NiPc/CNTs exhibited even higher selectivity for CO$_2$RR, with negligible H$_2$ detected (Fig. 3i). Therefore, the rate-limiting step on NiPc/CNTs is the protonation of adsorbed *CO$_2$, consistent with previous studies showing a high energy barrier for this step at the Ni−N$_4$ sites[20,38]. In addition, the KIE was evaluated for the concurrent HER during the CO$_2$ electrolysis. As shown in Supplementary Fig. 5, the H$_2$ partial current densities in NaHCO$_3$/H$_2$O were consistently higher than the corresponding D$_2$ partial current densities in NaDCO$_3$/D$_2$O for all catalysts, demonstrating the reliability of the KIE method in probing proton involvement in the rate-limiting step.

The above studies were performed in near-neutral electrolytes, while electrolyte pH may also influence the rate-limiting step of CO$_2$RR on metal phthalocyanines. Therefore, we further examined the rate-limiting step of CO$_2$RR on CoPc/CNTs in acidic media (1 M NaClO$_4$/H$_2$O and 1 M NaClO$_4$/D$_2$O, pH/pD = 2) using KIE analysis. As exhibited in

Supplementary Fig. 6, the partial current densities for CO production in NaClO$_4$/H$_2$O were consistently higher than those in NaClO$_4$/D$_2$O, yielding KIE values of 1.3–1.7. This indicates that the rate-limiting step of CO$_2$RR on metal phthalocyanines is not altered by pH variation. In addition, methanol can be formed as a liquid product during CO$_2$RR on CoPc/CNTs[31], which was detected in 1 M NaHCO$_3$ electrolyte at more negative potentials (<−0.80 V versus RHE), as shown in Supplementary Fig. 7. However, methanol formation occurs only at high overpotentials with relatively low selectivity compared to CO production. The possible divergence between our methanol production performance and prior reports likely arises from the high sensitivity of this pathway to experimental parameters, such as CoPc-to-CNT mass ratio, catalyst loading, and CO$_2$ partial pressure[31,33–35]. Given that the present work is centered on CO$_2$RR to CO on metal phthalocyanines and aims to elucidate the fundamental rate-limiting steps governing this primary reaction pathway, the following studies will focus on the CO$_2$-to-CO conversion.

## Effect of molecular dispersion state on the rate-limiting step

As previously discussed, the rate-limiting step of CO$_2$RR on CoPc catalyst remains controversial, with prior studies also identifying CO$_2$ adsorption as the rate-limiting step using the KIE method[37]. This discrepancy could stem from variations in CoPc immobilization methods, which lead to different dispersion states that have been reported to influence CO$_2$RR activity and selectivity[18,33]. For instance, the commonly used method of directly depositing CoPc dissolved in *N,N*-Dimethylformamide (DMF) onto substrates may cause aggregation of CoPc molecules due to the low surface area of substrates. To test this hypothesis, we prepared an additional CoPc electrode by directly depositing a DMF solution of CoPc onto an AvCarb GDS2230 carbon substrate. SEM characterization was then performed to examine its morphology. As shown in Supplementary Fig. 8, the electrode

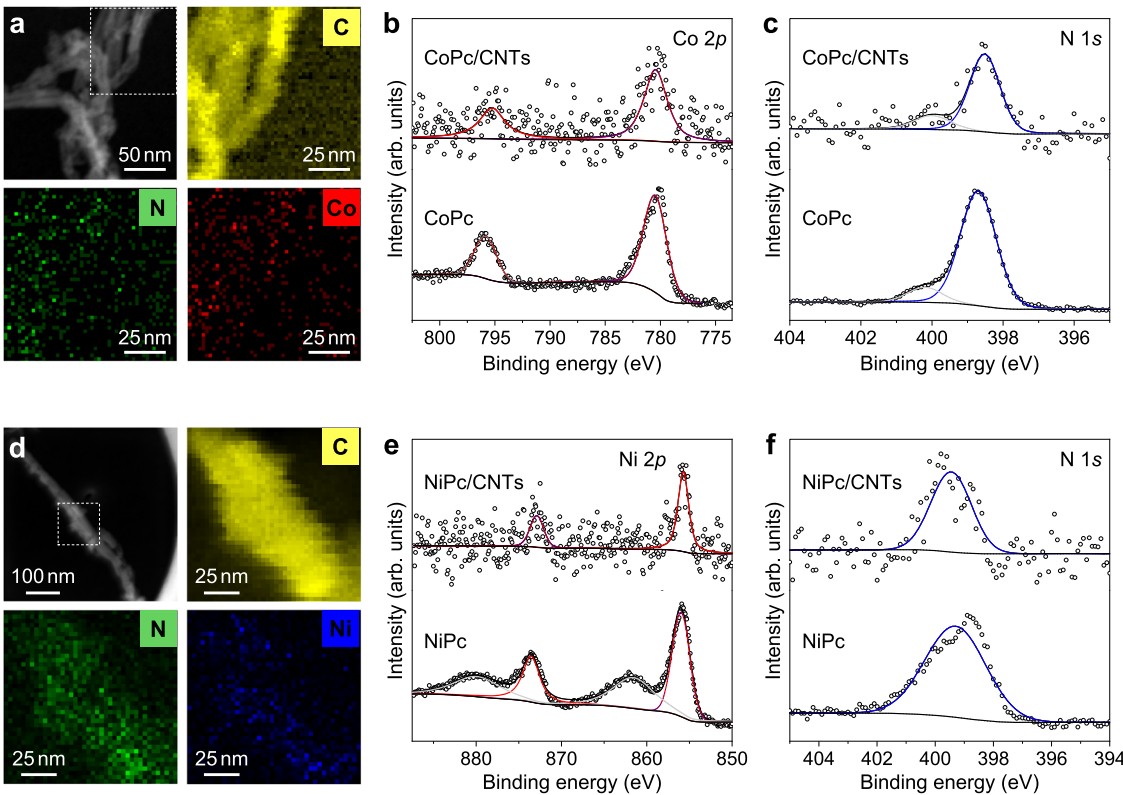

**Fig. 2 | Characterization of the CoPc/CNTs and NiPc/CNTs catalysts.** STEM image and corresponding EDS mapping (**a**), and high-resolution XPS spectra of Co 2*p* (**b**) and N 1*s* (**c**) of the CoPc/CNTs sample. STEM image and corresponding EDS mapping (**d**), and high-resolution XPS spectra of Ni 2*p* (**e**) and N 1*s* (**f**) of the NiPc/CNTs sample. Source data are provided as a Source data file.

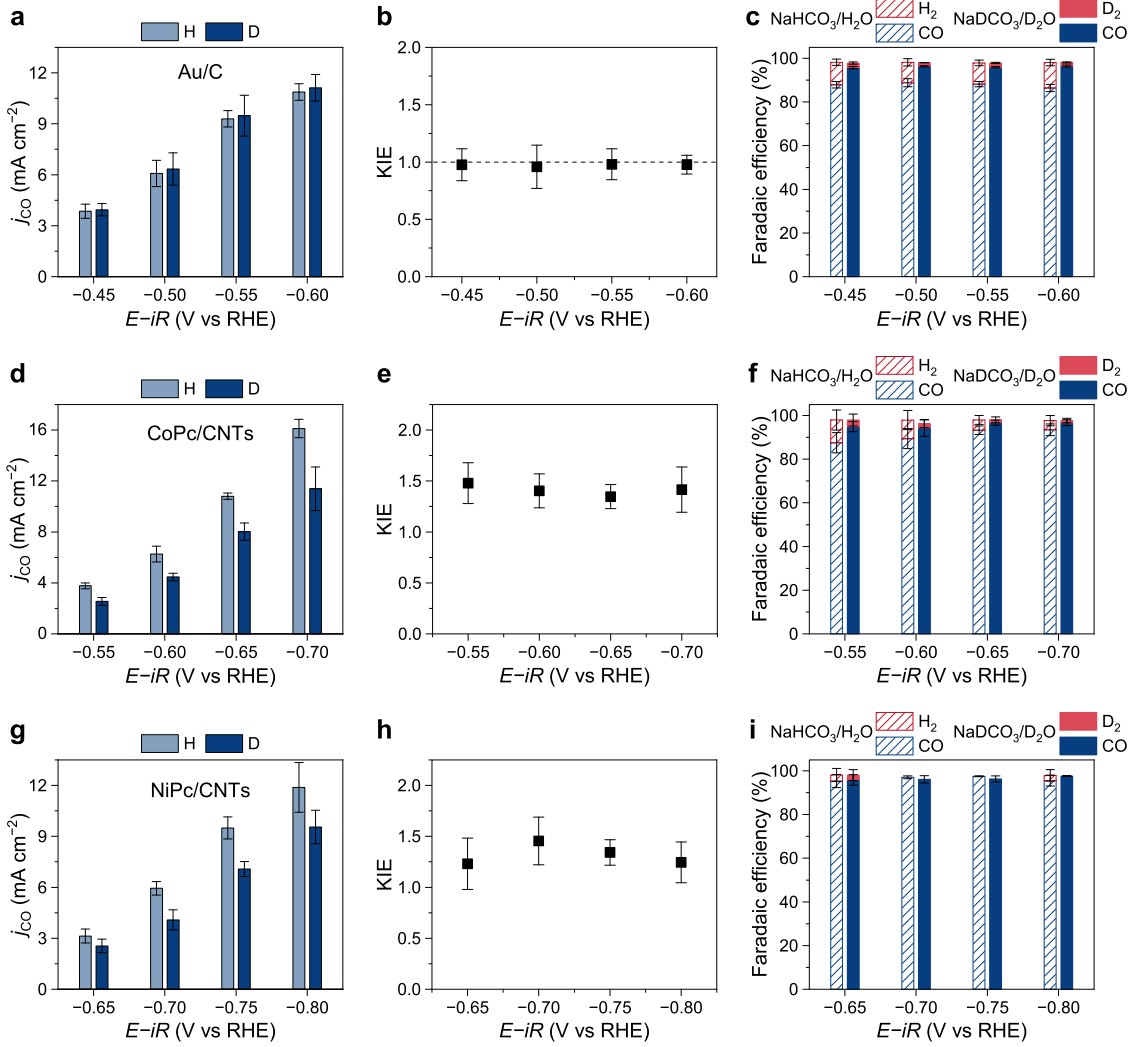

**Fig. 3 | KIE analysis of CO$_2$RR on Au/C, CoPc/CNTs, and NiPc/CNTs.** Partial current densities for CO production (**a**, **d**, **g**), corresponding KIE values (**b**, **e**, **h**), and Faradaic efficiencies (**c**, **f**, **i**) measured for the CO$_2$RR on the Au/C (**a**–**c**), CoPc/CNTs (**d**–**f**), and NiPc/CNTs (**g**–**i**) electrodes in 1 M NaHCO$_3$/H$_2$O and 1 M NaDCO$_3$/D$_2$O electrolytes. The applied potentials were 100% *iR*-compensated using the current-interrupt method (NaHCO$_3$/H$_2$O: $R_u = 6.9 \pm 0.3$ Ω, pH = 7.4; NaDCO$_3$/D$_2$O: $R_u = 8.1 \pm 0.7$ Ω, pD = 7.8; geometric electrode area = 0.5 cm$^2$). The error bars represent the standard deviation of three independent measurements. Source data are provided as a Source data file.

exhibited newly formed nanoparticles and nanoflakes, ranging from submicrometers to a few micrometers, on the substrate, suggesting the formation of CoPc aggregates. The aggregated CoPc electrode was then evaluated for CO$_2$RR, which showed a lower activity compared to the above-tested CoPc/CNTs electrode, as exhibited in Fig. 4a. More importantly, the CO partial current densities in NaHCO$_3$/H$_2$O and NaDCO$_3$/D$_2$O were nearly identical, resulting in KIE values of around 1, as shown in Fig. 4b. The CO Faradaic efficiencies were all greater than 90% (Fig. 4c), suggesting that CO$_2$RR was the dominant reaction. In contrast, the concurrent HER activities in the two electrolytes still showed a notable difference, yielding KIE values greater than 3 and indicating reliable KIE measurements (Supplementary Fig. 9). The results confirm that the rate-limiting step of CO$_2$RR on the aggregated CoPc electrode does not involve protons and should be the CO$_2$ adsorption step.

To further examine the possible influence of CNTs on the above-observed difference in the rate-limiting step, we prepared a control sample by mixing CoPc powders and CNTs, which will hereafter be referred to as the "CoPc/CNTs mixture" sample. As shown in Supplementary Fig. 10, SEM imaging revealed the formation of CoPc aggregates of a few micrometers in size, with their crystalline structure

confirmed by the XRD pattern. The CoPc/CNTs mixture sample was then deposited onto an AvCarb GDS2230 substrate to form an electrode and evaluated for the KIE in CO$_2$RR. As exhibited in Fig. 4d–f, the sample showed comparable CO$_2$RR performance, including the partial current density and Faradaic efficiency for CO production, and the similar KIE values (~1) indicate that the CO$_2$RR has the same rate-limiting step—CO$_2$ adsorption. Meanwhile, the concurrent HER showed a typical KIE on the CoPc/CNTs mixture electrode (Supplementary Fig. 11). Collectively, the differing rate-limiting steps of CO$_2$RR on CoPc molecules can be attributed to their distinct dispersion states, which may account for discrepancies among previous studies. This correlation likely arises from variations in the electric field around the Co centers in CoPc molecules, as will be further discussed below.

Prior to investigating the underlying mechanism, it is essential to establish that the observed differences in CO$_2$RR activity between NaHCO$_3$/H$_2$O and NaDCO$_3$/D$_2$O do not arise from extrinsic factors, such as differences in CO$_2$ solubility or the p$K_a$ values of H$_2$O versus D$_2$O. First, the molar solubilities of CO$_2$ in H$_2$O and D$_2$O at 25 °C and 1 atm are both ~33.7 mmol L$^{-1}$[41], which cannot account for the pronounced CO$_2$RR activity differences. Moreover, in 1 M NaHCO$_3$/H$_2$O and 1 M NaDCO$_3$/D$_2$O electrolytes, the equilibrium between dissolved

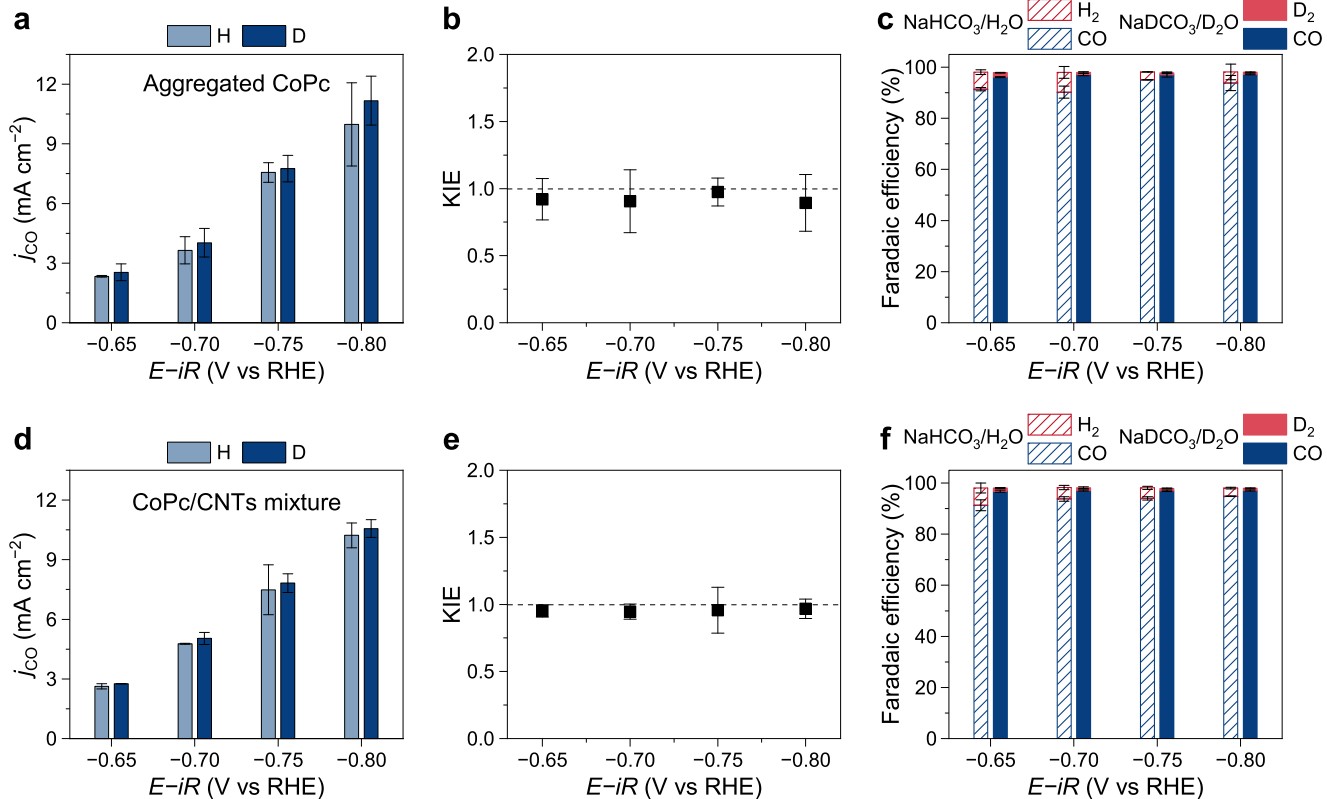

**Fig. 4 | Effect of molecular dispersion on the rate-limiting step of CO₂RR.** Partial current densities for CO production (**a**, **d**), corresponding KIE values (**b**, **e**), and Faradaic efficiencies (**c**, **f**) for CO₂RR on the aggregated CoPc electrode (**a**–**c**) and the CoPc/CNTs mixture electrode (**d**–**f**), measured in 1 M NaHCO₃/H₂O and 1 M NaDCO₃/D₂O electrolytes. The applied potentials were 100% *iR*-compensated using the current-interrupt method (NaHCO₃/H₂O: $R_u = 6.9 \pm 0.3\ \Omega$, pH = 7.4; NaDCO₃/ D₂O: $R_u = 8.1 \pm 0.7\ \Omega$, pD = 7.8; geometric electrode area = 0.5 cm²). The error bars represent the standard deviation of three independent measurements. Source data are provided as a Source data file.

CO₂ and bicarbonate species ($CO_2 + H_2O \rightleftharpoons H^+ + HCO_3^-$, $CO_2 + D_2O \rightleftharpoons D^+ + DCO_3^-$) can continuously buffer and replenish CO₂ near the cathode, ensuring a sustained and comparable CO₂ supply for the CO₂RR. Consequently, CO₂ solubility or availability can be excluded as the origin of the observed activity differences in the KIE studies.

Second, H₂O and D₂O exhibit a modest difference in their acid–base properties ($pK_w = 14.00$ for H₂O and 14.86 for D₂O), resulting in bulk pH/pD values of 7.4 and 7.8 for CO₂-saturated 1 M NaHCO₃ and NaDCO₃ electrolytes, respectively. In the above measurements, all potentials were converted to the RHE scale using the corresponding bulk pH, thereby accounting for pH-dependent thermodynamic effects in the comparison of CO₂RR activities. Nevertheless, local pH may change during reactions and influence reaction kinetics. To isolate genuine isotope effects from possible pH-induced artifacts, we employed in situ scanning electrochemical microscopy (SECM) to probe local pH under *operando* conditions. As shown in Supplementary Fig. 12, a customized SECM setup[42], equipped with an Au nanoelectrode tip whose exposed apex was functionalized with 4-hydroxylaminothiophenol (4-HATP)/4-nitrosothiophenol (4-NSTP), was used to measure the local pH near the cathode[43,44]. A correlation between the mid-peak potential of the 4-HATP/4-NSTP anodic peak and the electrolyte pH was established as a calibration curve (Supplementary Fig. 12). Using this approach, we quantified the local pH near the CoPc/CNTs electrode under various CO₂RR current densities, as presented in Supplementary Fig. 13. While a slight increase in local pH was observed during CO₂RR in both electrolytes, the relative difference between them remained ~0.4 pH units, consistent with the initial bulk pH difference (7.4 versus 7.8). Because this initial difference was already accounted for in the RHE-referenced potentials, the minor local pH shifts do not exert any additional influence on the KIE analysis.

## Mechanisms underlying distinct rate-limiting steps

The above electrochemical measurements and KIE analysis indicate that the rate-limiting step for CO₂RR to CO on CoPc molecules depends on their dispersion state. For molecularly dispersed CoPc on CNTs, the reaction is limited by the protonation of *CO₂, whereas for aggregated CoPc, CO₂ adsorption becomes the rate-limiting step. We hypothesize that the shift of the rate-limiting step on aggregated CoPc arises from a weakened interfacial electric field at the Co active sites. Bulk CoPc, as an organic semiconductor, exhibits low conductivity and pronounced dielectric behavior[45,46], which limits penetration of the applied electric field into thick CoPc aggregates[33]. Consequently, the Co active sites in aggregated CoPc experience a weaker electric field than those in molecularly dispersed CoPc, reducing the driving force for initial CO₂ activation and shifting the rate-limiting step. The change in the electric field at Co sites upon CoPc aggregation was revealed by Stark tuning, based on the potential-dependent frequency shifts of reaction intermediates in sum frequency generation spectra[33]. Here, we perform further mechanistic investigations to validate this hypothesis.

First, we compared the Co(II)/Co(I) redox responses of the CoPc/CNTs electrode and the CoPc/CNTs mixture electrode to examine the potential-screening effect due to CoPc aggregation. Because CNTs can introduce substantial capacitive currents and parasitic Faradaic contributions, we employed square-wave voltammetry (SWV), which can suppress the charging currents while providing high sensitivity[47,48]. A representative SWV waveform is illustrated in Fig. 5a. As shown in Fig. 5b, molecularly dispersed CoPc on CNTs experiences a strong local electric field, resulting in a pronounced and well-defined Co(II)/Co(I) redox peak at around 0 V vs RHE, consistent with previous reports[34]. In contrast, despite identical CoPc loadings, the redox peak of the

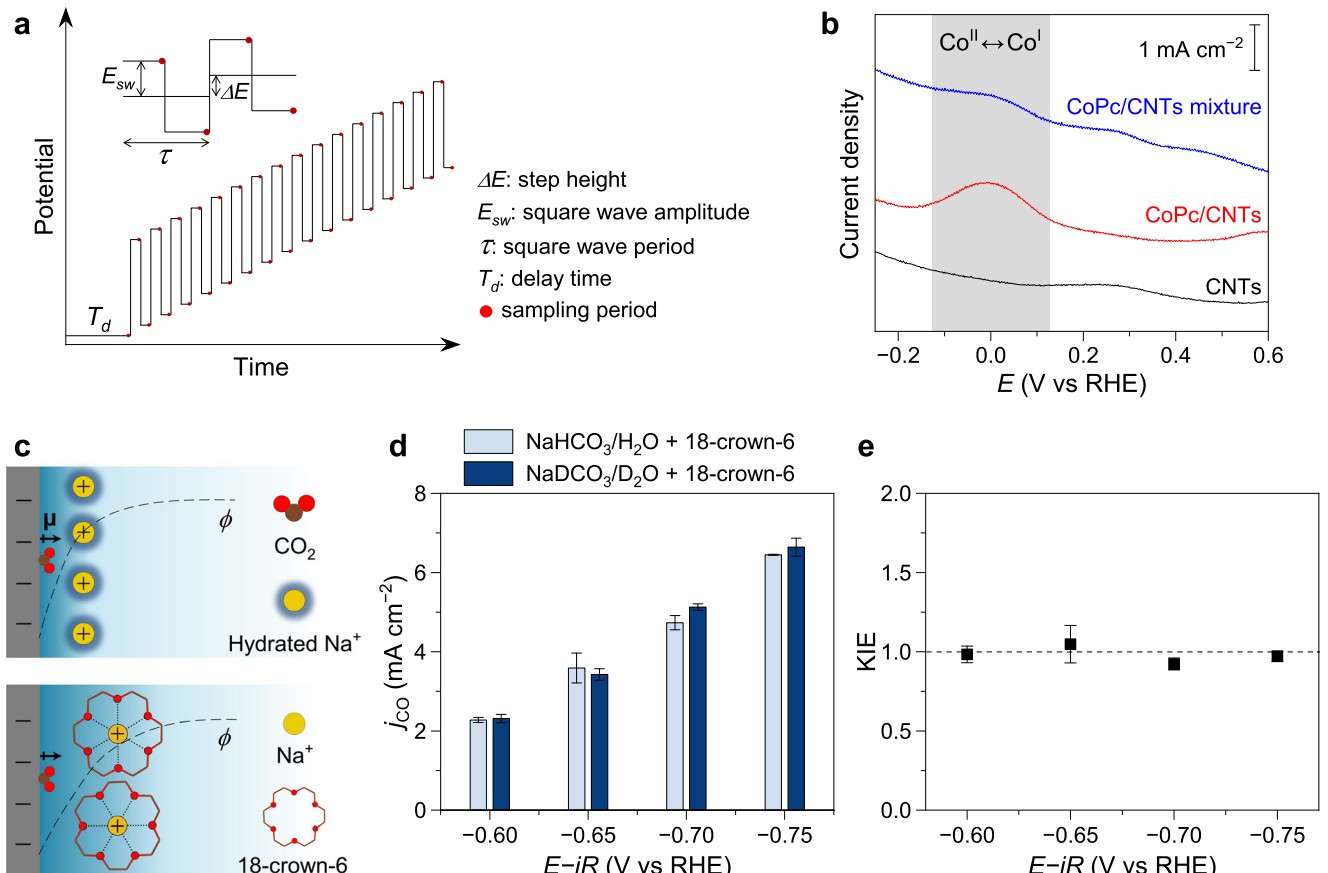

**Fig. 5 | Mechanistic insights into distinct rate-limiting steps. a** Schematic illustration showing the potential modulation in SWV. **b** SWV curves recorded on electrodes with different dispersion states of CoPc molecules. Step height: 2 mV; amplitude: 15 mV; frequency: 25 Hz. The applied potentials in (**b**) were not *iR*-compensated. **c** Schematic illustration showing the effect of 18-crown-6 on the interfacial electric field. **d** Partial current densities for CO production during $CO_2$RR on CoPc/CNTs in NaHCO$_3$/H$_2$O + 18-crown-6 and in NaDCO$_3$/D$_2$O + 18-crown-6

electrolytes. **e** Corresponding KIE values. The applied potentials in (**d**, **e**) were 100% *iR*-compensated using the current-interrupt method (NaHCO$_3$/H$_2$O + 18-crown-6: $R_u = 8.7 \pm 0.3\ \Omega$, pH = 7.4; NaDCO$_3$/D$_2$O + 18-crown-6: $R_u = 9.2 \pm 0.3\ \Omega$, pD = 7.8; geometric electrode area = 0.5 cm$^2$). The error bars represent the standard deviation of three independent measurements. Source data are provided as a Source data file.

aggregated CoPc is significantly attenuated compared to that of the molecularly dispersed CoPc, indicating a suppression of the Co(II)/Co(I) redox response due to aggregation. When a potential is applied to the electrode, the electric field does not fully penetrate the aggregates, limiting effective potential transmission to the Co active sites and resulting in a substantially attenuated redox signal.

To further examine whether attenuation of the electric field shifts the rate-limiting step to $CO_2$ adsorption, we intentionally reduced the local electric field around molecularly dispersed CoPc by chelating Na$^+$ in the electrolyte using the crown ether 18-crown-6[49]. Chelation of Na$^+$ effectively increases the apparent cation size and expands the thickness of the outer Helmholtz layer[50,51], thereby reducing the potential gradient and interfacial electric field, as schematically illustrated in Fig. 5c. KIE studies were then performed on the CoPc/CNTs electrode in 1 M NaHCO$_3$/H$_2$O + 18-crown-6 and in 1 M NaDCO$_3$/D$_2$O + 18-crown-6 electrolytes. As shown in Fig. 5d, the partial current densities for CO production greatly decreased in the NaHCO$_3$ + 18-crown-6 electrolyte compared to those in NaHCO$_3$ (Fig. 3d), indicating reduced activity due to the weakened electric field. Importantly, the $CO_2$RR activities in crown-ether-containing H$_2$O and D$_2$O electrolytes were nearly identical, giving KIE values of ~1 (Fig. 5e). These results indicate that upon weakening the electric field, the rate-limiting step shifts to $CO_2$ adsorption even for molecularly dispersed CoPc. In contrast, KIE values of ~3 were observed for the concurrent HER (Supplementary Fig. 14), confirming the reliability of KIE analysis in the presence of

crown ether. Overall, these findings demonstrate that the Co active sites in aggregated CoPc experience a weakened electric field so that $CO_2$ adsorption becomes rate-limiting. This provides a clear mechanistic explanation for the dependence of the $CO_2$RR rate-limiting step on the dispersion state of CoPc.

### Effect of electrolyte anions on the $CO_2$RR with CoPc/CNTs
The above insights into the rate-limiting step of $CO_2$RR on CoPc catalysts motivated an investigation of electrolyte anion effects, which have been relatively overlooked compared with cations. It is well-established that on typical metal catalysts, such as Au and Ag, $CO_2$RR relies on metal cations to facilitate the rate-limiting $CO_2$ adsorption by stabilizing adsorbed *$CO_2$ via electrostatic interaction, resulting in an apparent dependence on the cation identity and concentration[52–54]. In contrast, anions, including the widely used HCO$_3^-$, act more as a spectator without involvement in the rate-limiting step, as evidenced by the zero reaction order measured for $CO_2$RR on Au[55,56]. Given the distinct rate-limiting step on CoPc/CNTs, the effects of cations and anions on $CO_2$RR are expected to differ from those observed on metal catalysts. We thus evaluated the dependence of $CO_2$RR activity on the concentrations of Na$^+$ and HCO$_3^-$, respectively. To ensure a rigorous comparison, the concentration of HCO$_3^-$ was held constant while varying Na$^+$, and vice versa. Electrolyte electroneutrality was maintained by adding ClO$_4^-$ as necessary. As shown in Fig. 6a, the CO partial current density on CoPc/CNTs at −0.65 V versus RHE increased from

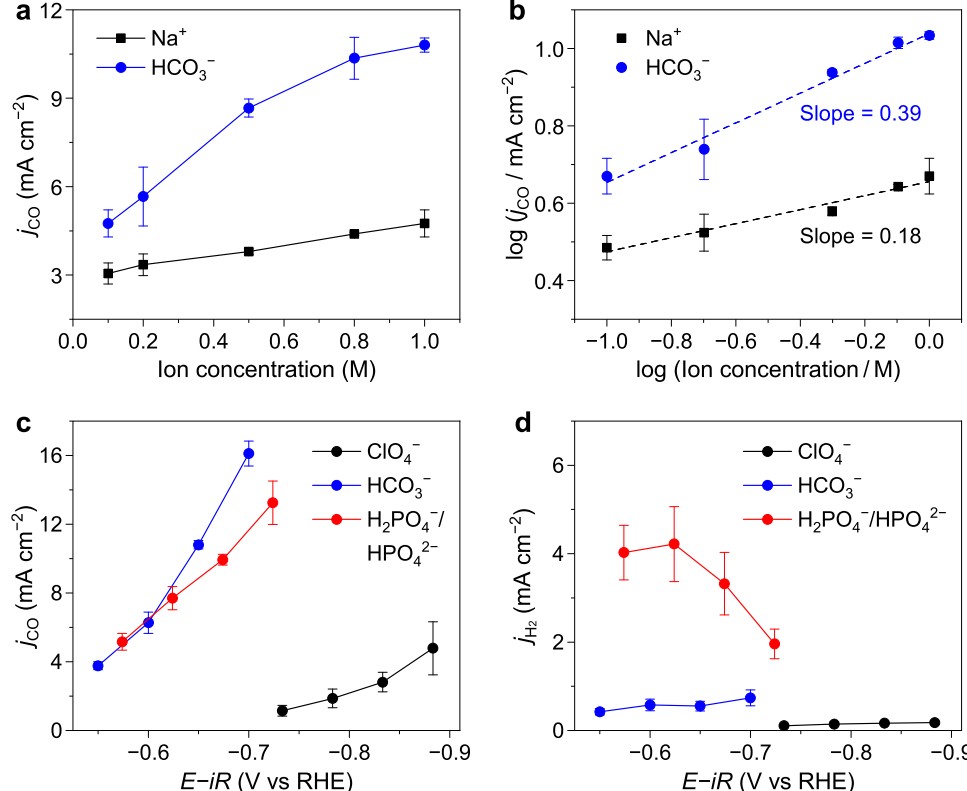

**Fig. 6 | Effect of electrolyte ions on $CO_2RR$ with CoPc/CNTs.** Partial current densities for CO production during $CO_2RR$ at −0.65 V versus RHE in electrolytes with varying concentrations of $Na^+$ and $HCO_3^-$ (**a**), and corresponding logarithmic plots (**b**) to determine reaction orders. Partial current densities for CO (**c**) and $H_2$ (**d**) production during $CO_2RR$ in electrolytes containing 1 M $Na^+$ paired with different anions. The applied potentials in (**c**, **d**) were 100% *iR*-compensated using the current-interrupt method ($ClO_4^-$: $R_u = 7.2 \pm 0.1\,\Omega$, pH = 4.3; $HCO_3^-$: $R_u = 6.9 \pm 0.3\,\Omega$, pH = 7.4; $H_2PO_4^-/HPO_4^{2-}$: $R_u = 11.6 \pm 0.4\,\Omega$, pH = 6.6; geometric electrode area = 0.5 cm²). The error bars represent the standard deviation of three independent measurements. Source data are provided as a Source data file.

3.1 to 4.7 mA cm⁻² as the $Na^+$ concentration increased from 0.1 to 1.0 M, yielding a reaction order of 0.18 (Fig. 6b). This cation effect likely arises from the promoted hydrolysis of the $Na^+$ hydration shell[32], which can facilitate the rate-limiting protonation step of adsorbed *$CO_2$. Interestingly, the $CO_2RR$ activity on CoPc/CNTs also increased with the $HCO_3^-$ concentration, even more pronounced than that for $Na^+$, resulting in a higher reaction order of 0.39. These results indicate that the $CO_2RR$ on CoPc/CNTs exhibits a stronger dependence on $HCO_3^-$ anion than on $Na^+$ cation, in sharp contrast to the behavior observed for metal catalysts[55,56]. This should be attributed to the promotional effect of $HCO_3^-$ on the rate-limiting protonation step of adsorbed *$CO_2$ on CoPc/CNTs, as $HCO_3^-$ can act as a proton donor in both $CO_2RR$ and HER, as reported in previous studies[57–59].

To further examine this effect, we selected anions with different proton-donating abilities, including $ClO_4^-$ (non-proton-donating), $HCO_3^-$ ($pK_a \sim 10.3$), and $H_2PO_4^-/HPO_4^{2-}$ ($pK_a \sim 7.2$), thus to compare their influence on the $CO_2RR$ with CoPc/CNTs. The comparative study was performed in electrolytes containing 1 M $Na^+$ paired with different anions. As shown in Fig. 6c, the $CO_2RR$ activity within the potential range of −0.55 to −0.90 versus RHE follows the trend: $ClO_4^- < H_2PO_4^-/HPO_4^{2-} \sim HCO_3^-$, reflecting differences in proton-donating ability, though not strictly mirroring their $pK_a$ values. This can be explained by including the concurrent HER activity during the electrolysis, which follows the trend: $ClO_4^- < HCO_3^- < H_2PO_4^-/HPO_4^{2-}$ (Fig. 6d). As a result, the $H_2PO_4^-/HPO_4^{2-}$ electrolyte showed the highest total current density (Supplementary Fig. 15), consistent with its stronger proton-donating capacity, although a larger fraction of the available protons was consumed by the HER, leading to lower Faradaic efficiencies for CO production. In addition, the $CO_2RR$ activity with $HCO_3^-$ became higher than that with $H_2PO_4^-/HPO_4^{2-}$ at more negative potentials

(Fig. 6c). This is attributed to the distinct role of $HCO_3^-$ as a $CO_2$ shuttle, which can increase the local concentration of $CO_2$ and the $CO_2RR$ activity[36,60], especially when the mass transport limit is approached at high reaction rates in H-cell. These results indicate that the $CO_2RR$ activity on CoPc/CNTs increases with the proton donation ability of electrolyte anions, confirming their role as a proton donor in the rate-limiting protonation step of adsorbed *$CO_2$.

The proton-donating ability of the anions may not only facilitate protonation steps in $CO_2RR$, but also provide buffering capacity to mitigate local pH changes during reactions[61,62]. To assess the contribution of such mechanistic factors, we employed the SECM with pH-sensitive probes to monitor the local pH during $CO_2RR$ in different electrolytes. As shown in Supplementary Fig. 16a, the local pH near the CoPc/CNTs electrode during $CO_2RR$ in $NaClO_4$ increased markedly, from an initial bulk pH of 4.3 to a local pH of 8.6 at a current density of 5.0 mA cm⁻², reflecting the absence of buffering capacity in $NaClO_4$. In contrast, the pH rise in $NaHCO_3$ was much smaller (Supplementary Fig. 13) due to its buffering nature. To account for pH-dependent thermodynamic effects when comparing $CO_2RR$ activities, we recalculated the activities using RHE-referenced potentials corrected by the measured local pH values (Supplementary Fig. 16b). Even after this correction, the $CO_2RR$ activities in $NaHCO_3$ remained consistently higher than those in $NaClO_4$, indicating that differences in buffering or local pH do not explain the variation in $CO_2RR$ activity among the anions. Instead, the higher $CO_2RR$ activity in $NaHCO_3$ is mainly attributed to enhanced *$CO_2$ protonation on CoPc/CNTs due to the proton-donating ability of the $HCO_3^-$ anion.

These findings suggest a strategy for enhancing $CO_2$ electrolysis systems with a rate-limiting protonation step through the use of proton-donating anions[61,62]. To further demonstrate this anion strategy

and its applicability, we evaluated $CO_2$ electrolysis in presence of different anions using a flow cell equipped with gas-diffusion electrodes (GDEs), where the $CO_2RR$ current density can be drastically increased due to enhanced mass transport[14]. Indeed, as shown in Supplementary Fig. 17, the $NaHCO_3$ electrolyte achieved the highest activity for $CO_2RR$ on CoPc/CNTs, followed by $NaH_2PO_4/Na_2HPO_4$ and $NaClO_4$. Moreover, given the negatively charged cathode during $CO_2RR$, we expect electrolyte cations capable of donating protons to be more effective in promoting the rate-limiting protonation step. Such cations can enter and accumulate in the electric double layer, allowing direct interactions with surface-bound intermediates and thereby enabling more efficient proton delivery to facilitate $*CO_2$ protonation.

## Discussion
The above studies focus on enhancing $CO_2RR$ on CoPc with a rate-limiting protonation step by leveraging the proton-donating ability of electrolyte ions. More broadly, the methodology demonstrated here, which identifies whether the $CO_2RR$ rate-limiting step involves proton transfer through KIE analysis and then optimizes the electrolyte accordingly, is applicable to other metal phthalocyanines with different metal centers (Ni, Fe, Cu, Mn) and to related molecular catalysts such as porphyrins and quaterpyridines. For catalysts with a protonation-limited pathway, $CO_2RR$ performance can be enhanced by selecting electrolyte ions with appropriate proton-donating ability. Conversely, for systems where $CO_2$ adsorption is rate-limiting, $CO_2RR$ activity can be improved by strengthening the interfacial electric field, for example, through the use of partially or weakly solvated cations[54]. Together, these insights provide generalizable guidance for the rational design of efficient $CO_2$ electrolysis systems.

In summary, we investigated the rate-limiting step of $CO_2RR$ to CO on metal phthalocyanines with Au as a reference. Using electrochemical measurements and KIE analysis, we identified the protonation of adsorbed $*CO_2$ as the rate-limiting step on molecularly dispersed CoPc and NiPc supported on CNTs. We further revealed that the rate-limiting step depends on the dispersion state of metal phthalocyanine molecules: on aggregated CoPc, it shifts to the $CO_2$ adsorption due to a weakened interfacial electric field at the Co active sites. These mechanistic insights also highlight a previously overlooked effect of electrolyte anions: $HCO_3^-$ strongly promotes the $CO_2RR$ on CoPc/CNTs by acting as a proton donor in the rate-limiting protonation step, as confirmed by a comparative study using anions with different proton donation abilities. Our study elucidates the rate-limiting step of $CO_2RR$ on metal phthalocyanine catalysts, establishes molecular dispersion as a key structural parameter, and provides guidance for the rational design of efficient molecular electrocatalysts.

## Methods
### Chemicals
Ethanol (99.5%, anhydrous, 200 proof), 2-propanol (HPLC), N,N-Dimethylformamide (DMF, Certified ACS), hydrochloric acid (HCl, 34–37%, trace-metal grade), sulfuric acid ($H_2SO_4$, 98%), cobalt (II) phthalocyanine (CoPc, $C_{32}H_{16}N_8Co$), tetrachloroauric acid trihydrate ($HAuCl_4 \cdot 3H_2O$, Certified ACS), sodium citrate ($C_6H_5Na_3O_7$, 99+%), tannic acid ($C_{76}H_{52}O_{46}$, Certified ACS), hydrogen peroxide ($H_2O_2$, 30%, Certified ACS), sodium carbonate ($Na_2CO_3$, 99.95%), sodium bicarbonate ($NaHCO_3$, 99.7+%), sodium perchlorate ($NaClO_4$, ACS Grade), sodium phosphate monobasic monohydrate ($NaH_2PO_4 \cdot H_2O$, 99+%), sodium phosphate dibasic heptahydrate ($Na_2HPO_4 \cdot 7H_2O$, 99+%), deuterium oxide ($D_2O$, 99.8 atom% D), titanium gauze (80 mesh), iridium(IV) chloride ($IrCl_4 \cdot xH_2O$), 18-Crown-6 (99%), and Au wires (0.5 mm diameter, 99.95%) were purchased from Fisher Scientific. Nickel (II) phthalocyanine (NiPc, $C_{32}H_{16}N_8Ni$), 4-nitrothiophenol (4-NTP), and Nafion perfluorinated resin solution (5 wt%) were purchased from Sigma Aldrich. Multi-walled carbon nanotubes (US4403, purity: >99.9 wt%, outside diameter: 10–20 nm) were purchased from US

Research Nanomaterials, Inc. Nafion 115 and 117 membranes, Vulcan XC-72 carbon black, and AvCarb GDS2230 carbon substrates were purchased from Fuel Cell Store. Ag/AgCl reference electrodes (MF-2056, 3 M KCl) were purchased from BASi Research Products. Leak-free Ag/AgCl reference electrodes (LF-1-100) were purchased from Innovative Instruments. Ar (99.999%), $H_2$ (99.999%), $CO_2$ (99.995%), and CO (99.9%) gases were purchased from Airgas. Deionized water with a specific resistance of 18.2 MΩ cm was used in all experiments.

### Preparation of catalysts and electrodes
The CoPc/CNTs sample was prepared using a method reported in previous studies[22,31]. Commercial CNTs were purified by heat treatment in air for 5 h, followed by sonication in a 5 wt% HCl solution for 30 min. The purified CNTs were collected by filtration, washed with deionized water, and dried at 60 °C. 50 mg of the purified CNTs and 1.5 mg of CoPc were each dispersed in 20 mL DMF and sonicated for 1 h. The two dispersions were then mixed, sonicated for 30 min, and stirred at room temperature (25 °C) for 20 h. The resulting solid was collected by centrifugation, washed with DMF and ethanol, and lyophilized to obtain the CoPc/CNTs sample. The CoPc content in the sample was determined to be 0.96 wt% using inductively coupled plasma optical emission spectroscopy (ICP-OES). The NiPc/CNTs sample was prepared using the same procedure as for CoPc/CNTs, with NiPc substituted for CoPc. The NiPc content in the NiPc/CNTs sample was determined to be 0.84 wt% using ICP-OES. To prepare the CoPc/CNTs mixture sample, 49.5 mg of purified CNTs and 0.5 mg CoPc were ground together in a motor for 30 min. Catalyst inks were prepared by mixing 5.6 mg of each sample (CoPc/CNTs, NiPc/CNTs, and the CoPc/CNTs mixture) with 1.6 mL of 2-propanol and 45 μL of Nafion solution (5 wt%), followed by sonication for 1 h. The inks were then drop-cast onto an AvCarb GDS2230 carbon substrate ($0.71 \times 0.71$ cm²; same dimensions for all electrodes used in the H-cell studies) until a loading of 0.5 mg cm⁻² (geometric area basis) was reached, resulting in the final electrodes. The aggregated CoPc electrode was prepared by drop-casting a DMF solution of CoPc with a concentration of 0.1 mg mL⁻¹ onto an AvCarb GDS2230 carbon substrate until a loading of 0.005 mg cm⁻² was reached. The CoPc loading on the CoPc/CNTs, CoPc/CNTs mixture, and aggregated CoPc electrodes was kept nearly identical for a rigorous comparison of their catalytic performances.

The Au/C nanopowder was prepared using a two-step method, including the synthesis of Au nanoparticles and subsequent loading onto carbon support. The Au nanoparticles were synthesized following the procedure described in a previous study[63]. 1 mL of tetrachloroauric acid ($HAuCl_4$, 25 mM) was injected into a mixed solution containing 150 mL of sodium citrate (2.2 mM), 0.1 mL of tannic acid (2.5 mM), and 1 mL of $Na_2CO_3$ (150 mM) at 50 °C. The resulting solid was collected by filtration, washed with deionized water, and freeze-dried to obtain the Au nanoparticles. 45 mg of Vulcan X-72 carbon black and 5 mg of the prepared Au nanoparticles were dispersed in 20 mL of deionized water and sonicated for 1 h. The dispersion was filtered, and the precipitate was washed with deionized water and dried at 60 °C overnight to obtain the Au/C sample. To prepare Au/C electrodes, 5.6 mg of the Au/C sample, 1.6 mL of 2-propanol, and 45 μL of Nafion solution (5 wt%) were mixed and sonicated for 1 h to form a catalyst ink. The ink was then drop-cast onto an AvCarb GDS2230 carbon substrate until a loading of 1.0 mg cm⁻² was reached.

### Materials characterization
SEM images were acquired using a ZEISS Ultra-55 FEG scanning electron microscope. TEM images, STEM images, and corresponding EDS mapping were acquired using a FEI Tecnai F30 transmission electron microscope with a field emission gun operated at 200 kV. XRD patterns were collected using a PANalytical Empyrean diffractometer with a 1.8 KW copper X-ray tube. XPS spectra were acquired using a Thermo

Scientific ESCALAB XI⁺ X-ray photoelectron spectrometer with an Al Kα X-ray source (1486.67 eV).

## Electrochemical measurements

All electrochemical measurements were performed using a Gamry Interface 1000 potentiostat with a gas-tight two-compartment electrochemical cell (H-cell, Supplementary Fig. 4) at room temperature (25 °C). The above prepared electrodes were used as cathodes, and an Ag/AgCl electrode and a piece of Pt gauze were used as the reference and counter electrodes, respectively. A Nafion 115 membrane (thickness: 127 μm) was used to separate the cathode and anode chambers. Prior to use, Nafion 115 membranes were sequentially heat-treated at 80 °C in 5% $H_2O_2$, 0.5 M $H_2SO_4$, and deionized water for 1 h each, then thoroughly rinsed and stored immersed in deionized water until use. The catholyte and anolyte each contained 15 mL of electrolyte. Four electrolyte compositions were used: 1 M $NaHCO_3$, 1 M $NaDCO_3$ in $D_2O$ (KIE experiments), 1 M $NaClO_4$, and a phosphate buffer (0.3 M $NaH_2PO_4$ + 0.35 M $Na_2HPO_4$). Electrolytes were prepared in $H_2O$ unless otherwise noted. The $NaDCO_3/D_2O$ electrolyte was prepared by bubbling $CO_2$ gas through a $Na_2CO_3$ solution in $D_2O$. Prior to $CO_2RR$ testing, the catholyte was purged with $CO_2$ for 30 min and then continuously purged at a rate of 5 sccm during bulk electrolysis while stirring at 600 rpm; the $CO_2$ gas flow rate was controlled and monitored using an Alicat Scientific mass flow controller. The GDE-cell studies were carried out using a home-built GDE flow cell[14], which consists of a Ti current collector with interdigitated gas-diffusion channels, a cathodic GDE with the catalyst layer deposited on AvCarb GDS2230 substrate, a 3D-printed chamber with ports for electrolyte flow and reference electrode, and an $IrO_x/Ti$ gauze anode inserted in a pocket of another Ti current collector. A leak-free Ag/AgCl electrode was used as the reference electrode. The catholyte and anolyte (15 mL each) were circulated using peristaltic pumps at 2.0 mL min⁻¹. All potentials were 100% iR-compensated using the current-interrupt method (unless otherwise specified) and are reported versus RHE. Potentials were measured versus an Ag/AgCl (3 M KCl) reference electrode and converted to the RHE scale using $E_{RHE} = E_{Ag/AgCl} + 0.210 V + 0.0591 × pH$. The reference electrode was calibrated using a standard hydrogen electrode before measurements. The reported current densities were normalized to the geometric surface areas of the electrodes. During $CO_2$ electrolysis, gas-phase products were quantified by a gas chromatograph (SRI Multiple Gas Analyzer #5) equipped with molecular sieve 5A and HayeSep D columns with Ar carrier gas. Liquid-phase products in the catholyte were analyzed by ¹H NMR spectroscopy (Bruker Avance III 500 MHz) after electrolysis.

## Calculation of CO₂RR activity and Faradaic efficiencies

The partial current densities for CO and $H_2$ production were calculated from the GC peak area as follows:

$$j_{partial} = \frac{\text{peak area}}{\alpha} \times (\text{gas flow rate}) \times \frac{nFp}{RT} \times (\text{electrode area})^{-1} \quad (1)$$

where $\alpha$ is a gas-dependent conversion factor determined by GC calibration, $n$ is the number of electrons transferred for the product, $F$ is the Faraday constant, $R$ is the ideal gas constant, $p$ and $T$ are the gas pressure and temperature, and the gas flow rate is the $CO_2$ feed rate delivered to the electrochemical cell (H-cell or GDE flow cell). The Faradaic efficiencies for CO and $H_2$ production were calculated by dividing the corresponding $j_{partial}$ by the total current density.

## Probing local pH using SECM

Local pH measurements were performed using a CH Instruments 920D SECM instrument equipped with three-directional stepper motor and piezoelectric actuators for coarse control and fine positioning of the SECM probe, respectively. A bipotentiostat was used to control the

potentials applied to the substrate working electrode (CoPc/CNTs on glassy carbon) and the SECM probe. The Au nanoelectrode SECM probe was fabricated by electrochemical etching of an Au wire to form a nanoscale apex radius, followed by electropolymerization coating to insulate the shaft while leaving the Au apex exposed[42]. The exposed Au apex was functionalized with a self-assembled monolayer of 4-nitrothiophenol (4-NTP), which was then electrochemically converted into the 4-HATP/4-NSTP redox couple[43,44]. The Au nanoelectrode design ensures that only the tip apex serves as the active sensing area, enabling local pH measurements with minimal perturbation to the reaction environment. To maintain consistent probe–substrate separation across measurements, the Au nanoelectrode probe was first brought to electrical contact with the substrate using the piezoelectric fine-approach routine and then retracted by 1 μm and held at that position during pH measurements under $CO_2RR$ conditions. Local pH was quantified by recording cyclic voltammograms at the Au nanoelectrode probe and monitoring the Nernstian shift of the mid-peak potential of the surface-bound 4-HATP/4-NSTP redox couple (Supplementary Fig. 12).

## Data availability

The data that support the findings of this study are available in the article and its Supplementary Information file. Source data are provided with this paper.

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

## Acknowledgements

This material is based upon work supported by the U.S. Department of Energy, Office of Science, Office of Basic Energy Sciences, Catalysis Science program under Award Number DE-SC0024083. Z.R. acknowledges the support from the Preeminent Postdoctoral Program (P3) at the University of Central Florida (UCF). Article processing charges were provided in part by the UCF College of Graduate Studies Open Access Publishing Fund.

## Author contributions

Z.R. and X.F. conceived and designed the study. Z.R. performed the experiments and analyzed the results with assistance from K.S., Z.M., and T.E. T.S.R. contributed to the discussion and interpretation of the results. Z.R. and X.F. wrote the manuscript with feedback from all authors.

## Competing interests

The authors declare no competing interests.
