## [Transparent Peer Review file · Nature Communications]

Elucidating the rate-limiting step of CO₂ electroreduction on metal phthalocyanines

Corresponding Author: Professor Xiaofeng Feng

Version 0:

Reviewer comments:

Reviewer #1

(Remarks to the Author)

This article by Ren et al. presents a mechanistic understanding on the rate-limiting step of CO₂ reduction over MPc catalysts. The dispersion state of MPc and electrolyte anions are considered to be important factors for the rate-limiting steps. KIE studies were used as the main method to determine whether the RDS involves a hydrogenation step. Yet, I feel that the manuscript is now too preliminary to be published. And, there are more issues regarding to the key points of MPc catalysts used in CO₂ electroreduction. The novelty and quality of this manuscript could not guarantee its publication in Nature Communications.

Additional comments below may help the authors to improve the manuscript.

1. The interesting point of MPc such as CoPc is that methanol can be produced as a major product when being loaded onto carbon nanotubes, rather than CO (refs.31-35). But it is not the case when being loaded on carbon black. I think revealing the underlying mechanism should make more sense.
2. The CoPc catalyst for both CO and methanol production has been well studied previously, even for the initial CO₂ to COOH step experimentally and theoretically, in spite of less KIE method. It is not surprised that the rate-limiting step over MPc is different from that over Au electrode.
3. With KIE experiments, the authors just demonstrate the possible RDS, but they failed to explain well the mechanism shift especially the role of dispersion on carbon support. The role of local electric field should be revealed with direct evidences, rather than discussing based on literature.
4. The role of anions including bicarbonate ions has been well documented in the literature. Further, it seems that the authors' logic just lies in the proton donating ability for a given hydrogenation step.
5. When conducting KIE studies, how did the authors exclude the effect of mass transport?

Reviewer #2

(Remarks to the Author)

This study provides valuable mechanistic insights into the dispersion-dependent rate-limiting steps of CO₂RR on CoPc catalysts, supported by electrochemical and KIE analyses. The identification of HCO₃⁻'s proton donor role in molecularly dispersed systems is particularly noteworthy. However, several critical issues require careful addressing to fully substantiate the conclusions. With these revisions, the work could offer a more robust foundation for molecular catalyst design. Consideration for publication would be appropriate after major revision.

Specific comments:

1. The KIE experiments conducted in NaHCO₃ and NaDCO₃ systems may be affected by the significant pK_a difference between D₂O and H₂O, leading to pH shifts that influence reaction kinetics. The authors should clarify how pH was controlled (e.g., buffer capacity adjustment or in situ monitoring) to isolate genuine isotope effects from pH-induced artifacts.
2. While SEM/XRD data characterize CoPc aggregation, claims about "electronic state changes" and "positioning farther from OHP" lack direct evidence. For example, XANES/EXAFS (electronic structure) and TOF-SIMS (interface distribution) should be added to support the proposed dispersion-dependent RLS mechanism.
3. The "proton donor" function of anions near the electrode interface may be constrained by electric double-layer structure (e.g., specific adsorption competition). A Gouy-Chapman-Stern model analysis is needed to distinguish intrinsic proton-

donating capacity from interfacial delivery efficiency.

4. The HCO₃⁻ proton donor mechanism depends not only on pK_a but also on local pH, buffer capacity, and applied potential. In situ techniques (such as SEIRAS, pH-sensitive probes) should be employed to decouple these intertwined factors.

5. The difference of CO₂ solubility in D₂O versus H₂O may artificially reduce current density in isotope experiments. Normalizing jCO values is essential for reliable KIE conclusions.

6. Anion comparisons in H-cells may not reflect practical electrolyzer conditions (flow cells, high current densities). Additional experiments are recommended to strengthen the applicability of the findings.

7. The focus on Co/Ni phthalocyanines lacks perspective on generalizability to other metal centers (Fe, Mn) or ligand frameworks (porphyrins, terpyridines). Expanding the discussion on design principles for broader applicability would enhance impact.

8. All potentials were reported against the SHE. It is intriguing to know how the experiment setup was constructed and how different pH impact was introduced in to the performance reporting.

9. Why was no liquid product detected from CoPc system?

10. Were the CoPc and NiPc molecules molecularly dispersed or forming aggregates on the CNT surface? Detailed characterization results should be supplied.

Reviewer #3

(Remarks to the Author)

This manuscript examines the reaction mechanism, specifically the rate-limiting step, of the electrochemical conversion of CO₂ to CO using metal phthalocyanine catalysts. The rate-limiting step is identified as the protonation of adsorbed *CO₂ on dispersed CoPc-supported CNTs. In contrast, on aggregated CoPc, CO₂ adsorption becomes the rate-limiting step. The authors investigated how the dispersion rate and electrolyte components (anions and cations) affect the CO₂RR to CO on CoPc/CNTs, thereby guiding the rational design of molecular electrocatalysts. Overall, this manuscript offers sufficient evidence to clarify the rate-determining step of CO₂RR to CO on metal phthalocyanine catalysts. Major Revision. Some specific comments are provided below.

1. I would recommend that authors add in-situ microscopy evidence to monitor the reaction intermediates to support the mechanism study further.

2. Except for the effects of anions and cations, will the pH variation affect the rate-limiting step on metal phthalocyanine?

Reviewer #4

(Remarks to the Author)

Version 1:

Reviewer comments:

Reviewer #1

(Remarks to the Author)

The authors have improved the quality of their manuscript by conducting new experiments and clarifying further mechanistic understanding. The local pH measurement by SECM is highly recommended. I am fine with most of their responses. Significant improvement can be seen in the revised manuscript compared to the original one.

Regarding to the product selectivity of CO versus methanol over CoPc with different supports, desperation and even Nafion additives, I would suggest the authors to provide more robust evidences, as this is an important unaddressed issue in the field. While the reported methanol FE in this work is low even at very negative potentials, higher methanol selectivity at less negative potentials is often reported in literature. Insights into such divergence would make much sense.

Reviewer #2

(Remarks to the Author)

The authors have provided comprehensive and thorough responses to the raised questions, supplemented with robust evidence. The manuscript has been substantially improved and is now recommended for publication.

Reviewer #3

(Remarks to the Author)

They have done an excellent work for the revision. I do not have extra comments this time. Please make a decision for this manuscript.

Reviewer #4

(Remarks to the Author)

Version 2:

Reviewer comments:

Reviewer #1

(Remarks to the Author)

The authors have addressed my comments. The current manuscript is now suitable for publication in Nature Communications.

Response to Referees

Reviewer #1:

This article by Ren et al. presents a mechanistic understanding on the rate-limiting step of CO₂ reduction over MPc catalysts. The dispersion state of MPc and electrolyte anions are considered to be important factors for the rate-limiting steps. KIE studies were used as the main method to determine whether the RDS involves a hydrogenation step. Yet, I feel that the manuscript is now too preliminary to be published. And, there are more issues regarding to the key points of MPc catalysts used in CO₂ electroreduction. The novelty and quality of this manuscript could not guarantee its publication in Nature Communications.

Response: We thank the reviewer for the constructive comments and suggestions, which have helped us improve the manuscript. In response, we have performed many new experiments with new data and discussions to deepen the mechanistic understanding and strengthen the manuscript. Our major revisions include:

- (1) We have gained a deeper understanding of the mechanism and attributed the shift in the rate-limiting step on aggregated CoPc to a weakened interfacial electric field at the Co active sites, as supported by solid evidence presented in Figure 5 of the revised manuscript.
- (2) We have employed *in situ* scanning electrochemical microscopy (SECM) to probe the local pH under *operando* conditions and have directly addressed all comments related to local pH, as demonstrated by the results in Supplementary Figs. 12, 13, and 16.
- (3) We have addressed your other comments by adding new results and discussions in the revised manuscript.

With the substantial improvements, our study now elucidates the previously debated rate-limiting step of CO₂RR on metal phthalocyanines, provides new mechanistic insights into the interfacial electric field and the local pH during CO₂RR, establishes molecular dispersion as a key structural parameter, and offers guidance for the rational design of efficient molecular electrocatalysts. These advances provide strong evidence for the novelty and quality of our revised manuscript.

Our point-to-point responses to the comments and corresponding revisions are described below. A copy of the revised manuscript with all changes highlighted using “Track Changes” function is provided to facilitate review.

Additional comments below may help the authors to improve the manuscript.

1. The interesting point of MPc such as CoPc is that methanol can be produced as a major product when being loaded onto carbon nanotubes, rather than CO (refs. 31-35). But it is not the case when being loaded on carbon black. I think revealing the underlying mechanism should make more sense.

Response: We thank the reviewer for this comment. First, we agree that methanol (CH₃OH) formation on CoPc is intriguing, as CH₃OH is a more reduced and valuable product than CO and is rarely obtained with high selectivity on other electrocatalysts. However, upon surveying the literature, we found pronounced variability in the reported CH₃OH production performance on CoPc. Even when CoPc is supported on carbon nanotubes (CNTs), the reported partial current densities for CH₃OH vary widely (from negligible to ~10 mA cm⁻²), and the corresponding Faradaic efficiencies span from negligible to ~40% (*Nature* **2019**, 575, 639–642; *Nat. Catal.* **2024**,

7, 1000–1009; *ACS Catal.* **2024**, *14*, 366–372; *ACS Appl. Energy Mater.* **2024**, *7*, 3091–3098; *J. Mater. Chem. A* **2024**, *12*, 31547–31556). Such large discrepancies should arise from the high sensitivity of CH₃OH formation to operational parameters, including mass ratio of CoPc to CNTs, total catalyst loading, CO₂ partial pressure, and even the amount of Nafion solution used in the catalyst ink (*J. Am. Chem. Soc.* **2024**, *146*, 16348–16354). These reports indicate that CH₃OH is not stably produced as a major product on CoPc/CNTs, especially when compared to CO.

Mechanistically, CO₂-to-CH₃OH on CoPc has been proposed to follow a relay pathway (CO₂ → CO → CH₃OH). In this context, the initially generated CO is expected to play a crucial role in enabling subsequent CH₃OH formation. Yet, the primary CO₂-to-CO step, and particularly its rate-limiting step on CoPc, remains debated. This knowledge gap has fundamentally limited the rational design of CoPc-based electrocatalytic system for efficient CH₃OH production. Therefore, our work that elucidates the rate-limiting step of CO₂-to-CO conversion on CoPc as well as other metal phthalocyanines is also essential for understanding and guiding CH₃OH production.

Second, the product selectivity of CO₂RR on CoPc is highly potential-dependent. Therefore, we evaluated CO₂RR in 1 M NaHCO₃ at more negative potentials to address the concern regarding CH₃OH production on CoPc/CNTs, as shown in the figure below.

CoPc/CNTs indeed produces CH₃OH, but only within a narrow and more negative potential window (< -0.80 V vs RHE), and with much lower Faradaic efficiencies compared to CO. Under our experimental conditions, which were not specifically optimized for CH₃OH formation, CH₃OH is not a major product for the CO₂RR on CoPc. We have added the above figure as Supplementary Fig. 7 with the following discussion added into the revised manuscript: “*In addition, methanol can be formed as a liquid product during CO₂RR on CoPc/CNTs,³¹ which was detected in 1 M NaHCO₃ electrolyte at more negative potentials (< -0.80 V versus RHE), as shown in Supplementary Fig. 7. However, methanol formation occurs only at high overpotentials with relatively low selectivity compared to CO production. Given that the present work is centered on CO₂RR to CO on metal phthalocyanines and aims to elucidate the fundamental rate-limiting steps governing this primary reaction pathway, the following studies will focus on the CO₂-to-CO conversion.*”

Additionally, in response to your comment regarding CoPc supported on carbon black, we tested and compared the CH₃OH production with CoPc supported on CNTs (CoPc/CNTs) and on carbon black (CoPc/CB) at representative potentials, as shown in the figures below.

CoPc/CB exhibits slightly lower activity but comparable selectivity for CH₃OH production relative to CoPc/CNTs. Thus, under our conditions, the support effect of CoPc molecules on methanol production is not apparent. The possible divergence from other reports can be attributed to the extreme sensitivity of CH₃OH production on CoPc to subtle variations in their experimental conditions. While this intriguing topic requires more dedicated and systematic investigation, such exploration is clearly beyond the scope of our present study.

2. The CoPc catalyst for both CO and methanol production has been well studied previously, even for the initial CO₂ to COOH step experimentally and theoretically, in spite of less KIE method. It is not surprised that the rate-limiting step over MPc is different from that over Au electrode.

Response: We thank the reviewer for this comment. We agree that CO₂RR on CoPc has been extensively investigated and that the rate-limiting step for CO production has been discussed in the literature. However, as discussed in the original manuscript, this crucial step **remains under debate**, with both CO₂ adsorption and *CO₂ protonation to *COOH proposed as possible rate-limiting steps in recent literature (*Nat. Commun.* **2019**, *10*, 1683; *Nat. Catal.* **2024**, *7*, 1000–1009; *Nat. Catal.* **2021**, *4*, 1024–1031; *Angew. Chem. Int. Ed.* **2024**, *63*, e202317942). Clarifying this ambiguity through reliable KIE analysis is therefore essential for defining a clear mechanistic basis for CO₂RR on CoPc. Moreover, as the initially produced CO serves as the reactant for downstream methanol formation, establishing the rate-limiting step for CO production also offers mechanistic insights relevant to the methanol pathway.

In our work, Au was employed as a **benchmark catalyst and reference** to demonstrate the validity of the KIE approach in identifying the CO₂RR rate-limiting step, rather than to highlight differences between metals and metal phthalocyanines. Our focus is on revealing the dispersion-dependent nature of the rate-limiting step on metal phthalocyanines, namely, the distinct behavior observed between molecularly dispersed and aggregated CoPc, which to our knowledge is reported here for the first time. Furthermore, the underlying mechanism for this dispersion dependence is further elucidated through additional mechanistic studies in the revised manuscript (as detailed in the following response).

Finally, we emphasize that distinguishing whether CO₂ adsorption or *CO₂ protonation is rate-limiting is of direct relevance for rational design of the system. For instance, a rate-limiting adsorption step would motivate strategies to strengthen the interfacial electric field, whereas a rate-limiting protonation step would benefit from the use of electrolytes with stronger proton-donating

ability. Therefore, we believe that the present study provides practical mechanistic guidance for optimizing CO₂RR performance on metal phthalocyanines.

3. With KIE experiments, the authors just demonstrate the possible RDS, but they failed to explain well the mechanism shift especially the role of dispersion on carbon support. The role of local electric field should be revealed with direct evidences, rather than discussing based on literature.

Response: We thank the reviewer for this comment, which prompted us to conduct an in-depth mechanistic investigation into the rate-limiting step shift induced by the dispersion state of CoPc. Following the reviewer's suggestion, we carefully designed and performed additional experiments to explicitly demonstrate that this shift arises from changes in the interfacial electric field at the Co active sites. **First**, we employed the Co(II)/Co(I) redox response of CoPc to evaluate the local electric field using the highly sensitive square-wave voltammetry. We observed a substantially attenuated redox peak for the aggregated CoPc relative to that of the molecularly dispersed CoPc, indicating a weakened electric field at Co active sites in aggregated CoPc, as shown in the figure below. **Next**, we examined whether attenuation of the electric field shifts the rate-limiting step to CO₂ adsorption. We intentionally reduced the local electric field around molecularly dispersed CoPc by expanding the thickness of the outer Helmholtz layer, achieved by increasing the effective size of Na⁺ using a crown ether. KIE experiments reveal that upon weakening the electric field, the rate-limiting step indeed shifts to CO₂ adsorption, even for molecularly dispersed CoPc. These results provide a clear mechanistic explanation that the dispersion-dependent shift in the CO₂RR rate-limiting step on CoPc arises from changes in the interfacial electric field, supported by **direct evidences** obtained from our carefully designed experiments.

The above figure has been included as Figure 5, and a comprehensive, in-depth discussion has been added to the revised manuscript as follows:

“Mechanisms underlying distinct rate-limiting steps. *The above electrochemical measurements and KIE analysis indicate that the rate-limiting step for CO₂RR to CO on CoPc molecules depends on their dispersion state. For molecularly dispersed CoPc on CNTs, the reaction is limited by the protonation of *CO₂, whereas for aggregated CoPc, CO₂ adsorption becomes the rate-limiting step. We hypothesize that the shift of the rate-limiting step on aggregated CoPc arises from a weakened interfacial electric field at the Co active sites. Bulk CoPc, as an organic semiconductor, exhibits low conductivity and pronounced dielectric behavior,^{45,46} which limits penetration of the applied electric field into thick CoPc aggregates.³³ Consequently, the Co active sites in aggregated CoPc experience a weaker electric field than those in molecularly dispersed CoPc, reducing the driving force for initial CO₂ activation and shifting the rate-limiting step. The change in the electric field at Co sites upon CoPc aggregation was revealed by Stark tuning, based on the potential-dependent frequency shifts of reaction intermediates in sum frequency generation spectra.³³ Here we perform further mechanistic investigations to verify the hypothesis.*

First, we compared the Co(II)/Co(I) redox responses of the CoPc/CNTs electrode and the CoPc/CNTs mixture electrode to examine the potential-screening effect due to CoPc aggregation. Because CNTs can introduce substantial capacitive currents and parasitic Faradaic contributions, we employed square-wave voltammetry (SWV), which can suppress the charging currents while providing high sensitivity.^{47,48} A representative SWV waveform is illustrated in Fig. 5a. As shown in Fig. 5b, molecularly dispersed CoPc on CNTs experiences a strong local electric field, resulting in a pronounced and well-defined Co(II)/Co(I) redox peak at around 0 V vs RHE, consistent with previous reports.³⁴ In contrast, despite identical CoPc loadings, the redox peak of the aggregated CoPc is significantly attenuated compared to that of the molecularly dispersed CoPc, indicating a suppression of the Co(II)/Co(I) redox response due to aggregation. When a potential is applied to the electrode, the electric field does not fully penetrate the aggregates, limiting effective potential transmission to the Co active sites and resulting in a substantially attenuated redox signal.

To further examine whether attenuation of the electric field shifts the rate-limiting step to CO₂ adsorption, we intentionally reduced the local electric field around molecularly dispersed CoPc by chelating Na⁺ in the electrolyte using the crown ether 18-crown-6.⁴⁹ Chelation of Na⁺ effectively increases the apparent cation size and expands the thickness of the outer Helmholtz layer,^{50,51} thereby reducing the potential gradient and interfacial electric field, as schematically illustrated in Fig. 5c. KIE studies were then performed on the CoPc/CNTs electrode in 1 M NaHCO₃/H₂O + 18-crown-6 and in 1 M NaDCO₃/D₂O + 18-crown-6 electrolytes. As shown in Fig. 5d, the partial current densities for CO production greatly decreased in the NaHCO₃ + 18-crown-6 electrolyte compared to those in NaHCO₃ (Fig. 3d), indicating reduced activity due to the weakened electric field. Importantly, the CO₂RR activities in crown-ether-containing H₂O and D₂O electrolytes were nearly identical, giving KIE values of ~1 (Fig. 5e). These results indicate that upon weakening the electric field, the rate-limiting step shifts to CO₂ adsorption even for molecularly dispersed CoPc. In contrast, KIE values of ~3 were observed for the concurrent HER (Supplementary Fig. 14), confirming the reliability of KIE analysis in the presence of crown ether. Overall, these findings demonstrate that the Co active sites in aggregated CoPc experience a weakened electric field so

that CO₂ adsorption becomes rate-limiting. This provides a clear mechanistic explanation for the dependence of the CO₂RR rate-limiting step on the dispersion state of CoPc.”

4. The role of anions including bicarbonate ions has been well documented in the literature. Further, it seems that the authors’ logic just lies in the proton donating ability for a given hydrogenation step.

Response: We thank the reviewer for this comment. We agree that the role of anions, particularly bicarbonate (HCO₃⁻), in CO₂RR has been well documented. However, most previous studies focused on metal catalysts (e.g., Au, Cu) and have identified two primary roles for HCO₃⁻: (i) a spectator role within kinetically controlled regimes (*J. Am. Chem. Soc.* **2017**, *139*, 17109–17113; *Acc. Chem. Res.* **2022**, *55*, 1900–1911), and (ii) a CO₂-shuttle role when mass-transport limitation is approached (*ACS Energy Lett.* **2018**, *3*, 722–723; *J. Am. Chem. Soc.* **2015**, *137*, 4701–4708). The spectator role is consistent with a rate-limiting CO₂ adsorption step on metal catalysts, which is largely unaffected by HCO₃⁻, and the CO₂-shuttle behavior reflects the capability of HCO₃⁻ replenishing CO₂ near the electrode under high reaction rates via the HCO₃⁻/CO₂ equilibrium.

By contrast, the role of HCO₃⁻ as a proton donor that participates in a rate-limiting protonation step of CO₂RR has been rarely reported. In our work we measured a reaction order of ~0.39 with respect to HCO₃⁻ for CO₂RR activity on CoPc/CNTs, which indicates a substantial promotional effect of HCO₃⁻. In addition, we further observed significantly higher CO₂RR activity in HCO₃⁻ or H₂PO₄⁻/HPO₄²⁻ electrolytes compared to ClO₄⁻. These results attribute the promotional effect of HCO₃⁻ to enhanced proton donation in the rate-limiting protonation of *CO₂ on CoPc/CNTs. Elucidating the distinct roles of HCO₃⁻ in CO₂RR on electrocatalysts with different rate-limiting steps is critical, as it reveals the non-negligible impact of electrolyte ions on reaction kinetics and guides the rational design of efficient CO₂ electrolysis systems.

Furthermore, to address the reviewer’s concern regarding other possible roles of HCO₃⁻ in CO₂RR, we further evaluated its buffering effect by comparing the local pH during CO₂RR on CoPc/CNTs in CO₂-saturated 1 M NaHCO₃ and in 1 M NaClO₄. To probe local pH under operando conditions, we employed *in situ* scanning electrochemical microscopy (SECM), as illustrated in figure below. A customized SECM setup developed in our own group (*Ph.D. Dissertation*, **2023**, University of Central Florida, Orlando, FL, <https://stars.library.ucf.edu/etd2020/1631/>) with an Au nanoelectrode tip functionalized with 4-hydroxylaminothiophenol (4-HATP)/4-nitrosothiophenol (4-NSTP) (*Anal. Chem.* **2020**, *92*, 2237–2243; *JACS Au* **2021**, *1*, 1915–1924) was used to measure the local pH in the vicinity of the cathode. A correlation between the mid-peak potential of the 4-HATP/4-NSTP anodic peak and the electrolyte pH was established as a calibration curve.

Using this method, we quantified the local pH at different CO_2RR current densities. As shown in the figure below, the local pH during CO_2RR in NaClO_4 increased substantially, from an initial bulk pH of 4.3 to 8.6 at a current density of 5.0 mA cm^{-2} , reflecting the absence of buffering capacity in NaClO_4 . In contrast, the pH change in NaHCO_3 was much smaller ($\sim 0.4 \text{ pH unit}$) due to its buffering nature.

More importantly, to examine whether this pH-buffering capability could explain the variation in CO_2RR activity among the anions, we compared the CO_2RR activities in the two electrolytes under RHE-referenced potentials corrected by the measured local pH values, thereby eliminating pH-dependent thermodynamic contributions. Even after this correction, the CO_2RR activity in NaHCO_3 remained consistently higher than that in NaClO_4 , as shown in (c) of the above figure. This remaining enhancement implies that the effect of HCO_3^- is not solely a buffering phenomenon, but includes an intrinsic proton-donating contribution that facilitates the rate-limiting protonation of $^*\text{CO}_2$ on CoPc/CNTs .

The above two figures have been incorporated into the revised manuscript, and the following discussion has been added: “*The proton-donating ability of the anions may not only facilitate protonation steps in CO₂RR, but also provide buffering capacity to mitigate local pH changes during reactions.^{61,62} To assess the contribution of such mechanistic factors, we employed the SECM with pH-sensitive probes to monitor the local pH during CO₂RR in different electrolytes. As shown in Supplementary Fig. 16a, the local pH near the CoPc/CNTs electrode during CO₂RR in NaClO₄ increased markedly, from an initial bulk pH of 4.3 to a local pH of 8.6 at a current density of 5.0 mA cm⁻², reflecting the absence of buffering capacity in NaClO₄. In contrast, the pH rise in NaHCO₃ was much smaller (Supplementary Fig. 13) due to its buffering nature. To account for pH-dependent thermodynamic effects when comparing CO₂RR activities, we re-calculated the activities using RHE-referenced potentials corrected by the measured local pH values (Supplementary Fig. 16b). Even after this correction, the CO₂RR activities in NaHCO₃ remained consistently higher than those in NaClO₄, indicating that differences in buffering or local pH do not explain the variation in CO₂RR activity among the anions. Instead, the higher CO₂RR activity in NaHCO₃ is mainly attributed to enhanced *CO₂ protonation on CoPc/CNTs due to the proton-donating ability of the HCO₃⁻ anion.*”

5. When conducting KIE studies, how did the authors exclude the effect of mass transport?

Response: We thank the reviewer for this valuable comment. We agree that excluding the influence of CO₂ mass transport is essential for a reliable interpretation of the KIE results and the associated rate-limiting step of CO₂RR. In our study, the influence of mass transport is minimized in the KIE studies for three reasons. **First**, CO₂ has essentially identical molar solubility in H₂O and D₂O, and the solubility of CO₂ at 25 °C and 1 atm is ~0.0337 mol L⁻¹ in both H₂O and D₂O (*J. Am. Chem. Soc.* 1938, 60, 2771–2773). Some literature reports CO₂ solubility on a mol kg⁻¹ basis, which introduces a slight difference between H₂O and D₂O, but this difference arises solely from the higher density of D₂O. When expressed on a mol L⁻¹ basis, which is the relevant metric for mass-transport considerations, the CO₂ solubility is essentially identical in the two solvents. **Second**, all KIE measurements in our study were performed in 1 M bicarbonate electrolytes, where the rapid equilibrium between dissolved CO₂ and bicarbonate species (CO₂ + H₂O ⇌ H⁺ + HCO₃⁻, CO₂ + D₂O ⇌ D⁺ + DCO₃⁻) continuously buffers and replenishes CO₂ near the electrode. This equilibrium ensures a sustained and comparable CO₂ supply during CO₂RR in both H₂O and D₂O electrolytes. **Finally**, the KIE measurements were conducted at relatively low current densities and low overpotentials, conditions under which reaction rates are dominated by intrinsic kinetics rather than mass transport.

Furthermore, the negligible influence of CO₂ mass transport to the KIE analysis is supported by the similar CO₂RR activities observed on Au in NaHCO₃/H₂O and NaDCO₃/D₂O (as shown in the figure below, from Figure 3 of our manuscript). Since CO₂ adsorption is the rate-limiting step on Au and thus highly sensitive to local CO₂ concentration, the comparable activities confirm that mass transport does not account for the observed activity differences in our KIE studies on metal phthalocyanines. Collectively, these results demonstrate that CO₂ mass transport can be excluded as the origin of the observed activity differences in the KIE studies.

The following discussion has been added to the revised manuscript: “Prior to investigating the underlying mechanism, it is essential to establish that the observed differences in CO₂RR activity between NaHCO₃/H₂O and NaDCO₃/D₂O do not arise from extrinsic factors, such as differences in CO₂ solubility or the pK_a values of H₂O versus D₂O. First, the molar solubilities of CO₂ in H₂O and D₂O at 25 °C and 1 atm are both ~33.7 mmol L⁻¹,⁴¹ which cannot account for the pronounced CO₂RR activity differences. Moreover, in 1 M NaHCO₃/H₂O and 1 M NaDCO₃/D₂O electrolytes, the equilibrium between dissolved CO₂ and bicarbonate species (CO₂ + H₂O ⇌ H⁺ + HCO₃⁻, CO₂ + D₂O ⇌ D⁺ + DCO₃⁻) can continuously buffer and replenish CO₂ near the cathode, ensuring a sustained and comparable CO₂ supply for the CO₂RR. Consequently, CO₂ solubility or availability can be excluded as the origin of the observed activity differences in the KIE studies.”

Reviewer #2:

This study provides valuable mechanistic insights into the dispersion-dependent rate-limiting steps of CO₂RR on CoPc catalysts, supported by electrochemical and KIE analyses. The identification of HCO₃⁻ as proton donor role in molecularly dispersed systems is particularly noteworthy. However, several critical issues require careful addressing to fully substantiate the conclusions. With these revisions, the work could offer a more robust foundation for molecular catalyst design. Consideration for publication would be appropriate after major revision.

Response: We thank the reviewer for the positive evaluation of our work and we deeply appreciate the constructive comments and suggestions, which have helped us greatly improve the manuscript. In response, we have performed many new experiments with new data and discussions to deepen the mechanistic understanding and strengthen the manuscript. Our major revisions include:

- (1) We have gained a deeper understanding of the mechanism and attributed the shift in the rate-limiting step on aggregated CoPc to a weakened interfacial electric field at the Co active sites, as supported by solid evidence presented in Figure 5 of the revised manuscript.
- (2) We have employed *in situ* scanning electrochemical microscopy (SECM) to probe the local pH under *operando* conditions and have directly addressed all comments related to local pH, as demonstrated by the results in Supplementary Figs. 12, 13, and 16.
- (3) We have revised the CO₂RR performance reporting from the SHE scale to RHE-referenced potentials in the revised manuscript, thus accounting for pH-dependent thermodynamic effects in the comparison of CO₂RR activities.
- (4) We conducted experiments using a GDE flow cell to compare the effects of different anions under practical electrolyzer conditions (Supplementary Fig. 17) to show the applicability.
- (5) We have addressed other remaining minor comments by adding new results and discussions in the revised manuscript.

Our point-to-point responses to the comments and corresponding revisions are described below. We believe that our manuscript has been greatly improved after the revisions. A copy of the revised manuscript with all changes highlighted using “Track Changes” is provided to facilitate review.

Specific comments:

1. The KIE experiments conducted in NaHCO₃ and NaDCO₃ systems may be affected by the significant pK_a difference between D₂O and H₂O, leading to pH shifts that influence reaction kinetics. The authors should clarify how pH was controlled (e.g., buffer capacity adjustment or *in situ* monitoring) to isolate genuine isotope effects from pH-induced artifacts.

Response: We thank the reviewer for this constructive comment. We agree that isolating intrinsic isotope effects from pH-induced artifacts is essential for a reliable interpretation of the KIE results and the associated rate-limiting step of CO₂RR. Following the reviewer’s suggestion, we have minimized pH-dependent thermodynamic contributions by: (i) reporting the CO₂RR activities in NaHCO₃/H₂O and NaDCO₃/D₂O on the RHE scale using their respective bulk pH values, and (ii) quantifying local pH shifts under *operando* conditions in both electrolytes using *in situ* scanning electrochemical microscopy (SECM).

First, we re-compared the CO₂RR performance in NaHCO₃/H₂O and NaDCO₃/D₂O on the RHE scale, rather than on the originally used SHE scale, to eliminate the initial thermodynamic

offset arising from the bulk pH difference between the two electrolytes. Accordingly, we repeated CO₂RR measurements on all catalysts (Au/C, CoPc/CNTs, NiPc/CNTs, the aggregated CoPc, and the CoPc/CNTs mixture) in CO₂-saturated 1 M NaDCO₃/D₂O while applying a negative potential shift of 24 mV relative to the NaHCO₃/H₂O condition. This correction corresponds to $-0.0591 \times \Delta\text{pH}$ (D₂O – H₂O), where $\Delta\text{pH} = 0.4$ based on the bulk pH values of 7.4 and 7.8 for CO₂-saturated 1 M NaHCO₃/H₂O and NaDCO₃/D₂O, respectively. The improved results are exhibited below, and the corresponding figures have been updated in the revised manuscript.

New Figure 3 in the revised manuscript:

New Figure 4 in the revised manuscript:

As presented above, the observed differences in CO₂RR activity between NaHCO₃/H₂O and Na₂CO₃/D₂O for molecularly dispersed CoPc/CNTs and NiPc/CNTs remain after applying the pH-corrected potential shifts, although the magnitude of the KIE values is slightly changed. This persistence indicates that the measured KIE primarily arises from an intrinsic isotope effect rather than from pH-induced thermodynamic artifacts, thereby confirming a rate-limiting protonation of *CO₂ on the CoPc/CNTs and NiPc/CNTs catalysts.

Second, considering that local pH can shift from bulk pH during CO₂RR, possibly introducing additional thermodynamic shifts that affect kinetics, we further used SECM to probe the local pH under operando conditions in the NaHCO₃/H₂O and Na₂CO₃/D₂O electrolytes, as shown in the figure below. A customized SECM setup developed in our own group (*Ph.D. Dissertation*, 2023, University of Central Florida, Orlando, FL, <https://stars.library.ucf.edu/etd2020/1631/>) with an Au nanoelectrode tip functionalized with 4-hydroxylaminothiophenol (4-HATP)/4-nitrosothiophenol (4-NSTP) was used to measure the local pH in the vicinity of the cathode (*Anal. Chem.* 2020, 92, 2237–2243; *JACS Au* 2021, 1, 1915–1924). A correlation between the mid-peak potential of the 4-HATP/4-NSTP anodic peak and the electrolyte pH was established as a calibration curve.

Using this approach, we successfully measured the local pH near the CoPc/CNTs electrode under various CO_2RR current densities, as presented in the figure below. While a slight increase in local pH was observed during CO_2RR in both electrolytes, the relative difference between them remained ~ 0.4 pH units, consistent with the initial bulk pH difference (7.4 versus 7.8). Because this initial difference was already accounted for in the RHE-referenced potentials, the minor local pH shifts do not exert any additional influence on the KIE analysis.

The above two figures has been added as Supplementary Figure 12 and Supplementary Figure 13, respectively. The following discussion has been incorporated into the revised manuscript: “Second, H_2O and D_2O exhibit a modest difference in their acid–base properties ($pK_w = 14.00$ for H_2O and 14.86 for D_2O), resulting in bulk pH/pD values of 7.4 and 7.8 for CO_2 -saturated 1 M NaHCO_3 and NaDCO_3 electrolytes, respectively. In the above measurements, all potentials were converted to the RHE scale using the corresponding bulk pH, thereby accounting for pH-dependent thermodynamic effects in the comparison of CO_2RR activities. Nevertheless, local pH may change during reactions and influence reaction kinetics. To isolate genuine isotope effects from possible pH-induced artifacts, we employed in situ scanning electrochemical microscopy (SECM) to probe local pH under operando conditions. As shown in Supplementary Fig. 12, a customized SECM setup,⁴² equipped with an Au nanoelectrode tip whose exposed apex was

functionalized with 4-hydroxylaminothiophenol (4-HATP)/4-nitrosothiophenol (4-NSTP), was used to measure the local pH near the cathode.^{43,44} A correlation between the mid-peak potential of the 4-HATP/4-NSTP anodic peak and the electrolyte pH was established as a calibration curve (Supplementary Fig. 12). Using this approach, we quantified the local pH near the CoPc/CNTs electrode under various CO₂RR current densities, as presented in Supplementary Fig. 13. While a slight increase in local pH was observed during CO₂RR in both electrolytes, the relative difference between them remained ~0.4 pH units, consistent with the initial bulk pH difference (7.4 versus 7.8). Because this initial difference was already accounted for in the RHE-referenced potentials, the minor local pH shifts do not exert any additional influence on the KIE analysis.”

2. While SEM/XRD data characterize CoPc aggregation, claims about "electronic state changes" and "positioning farther from OHP" lack direct evidence. For example, XANES/EXAFS (electronic structure) and TOF-SIMS (interface distribution) should be added to support the proposed dispersion-dependent RLS mechanism.

Response: We thank the reviewer for this comment, which prompted us to conduct an in-depth mechanistic investigation into the rate-limiting step shift induced by the dispersion state of CoPc. Following the reviewer’s suggestion, we designed and performed new experiments that explicitly demonstrate that this shift arises from changes in the interfacial electric field at the Co active sites. Notably, the change in the electric field at Co sites upon CoPc aggregation has been revealed in previous studies using Stark tuning, based on the potential-dependent frequency shifts of reaction intermediates in sum frequency generation spectra (*Nat. Catal.* **2024**, 7, 987–999). In this work, we conducted additional mechanistic studies to provide further evidence for this hypothesis.

First, we employed the Co(II)/Co(I) redox response of CoPc to evaluate the local electrical field using the highly sensitive square-wave voltammetry. We observed a substantially attenuated redox peak for the aggregated CoPc relative to that of molecularly dispersed CoPc, indicating a weakened electric field at Co active sites in aggregated CoPc, as shown in the figure below.

Next, we examined whether attenuation of the electric field shifts the rate-limiting step to CO₂ adsorption. We intentionally reduced the local electric field around molecularly dispersed CoPc by expanding the thickness of the outer Helmholtz layer, achieved by increasing the effective size of Na⁺ using a crown ether. KIE experiments reveal that upon weakening the electric field, the rate-limiting step indeed shifts to CO₂ adsorption, even for molecularly dispersed CoPc. These results provide a clear mechanistic explanation that the CoPc dispersion-dependent shift in the CO₂RR rate-limiting step arises from changes in the interfacial electric field.

The above figure has been added as Figure 5 and a comprehensive, in-depth discussion has been added to the revised manuscript as follows:

*“Mechanisms underlying distinct rate-limiting steps. The above electrochemical measurements and KIE analysis indicate that the rate-limiting step for CO₂RR to CO on CoPc molecules depends on their dispersion state. For molecularly dispersed CoPc on CNTs, the reaction is limited by the protonation of *CO₂, whereas for aggregated CoPc, CO₂ adsorption becomes the rate-limiting step. We hypothesize that the shift of the rate-limiting step on aggregated CoPc arises from a weakened interfacial electric field at the Co active sites. Bulk CoPc, as an organic semiconductor, exhibits low conductivity and pronounced dielectric behavior,^{45,46} which limits penetration of the applied electric field into thick CoPc aggregates.³³ Consequently, the Co active sites in aggregated CoPc experience a weaker electric field than those in molecularly dispersed CoPc, reducing the driving force for initial CO₂ activation and shifting the rate-limiting step. The change in the electric field at Co sites upon CoPc aggregation was revealed by Stark tuning, based on the potential-dependent frequency shifts of reaction intermediates in sum frequency generation spectra.³³ Here we perform further mechanistic investigations to verify the hypothesis.*

First, we compared the Co(II)/Co(I) redox responses of the CoPc/CNTs electrode and the CoPc/CNTs mixture electrode to examine the potential-screening effect due to CoPc aggregation. Because CNTs can introduce substantial capacitive currents and parasitic Faradaic contributions, we employed square-wave voltammetry (SWV), which can suppress the charging currents while providing high sensitivity.^{47,48} A representative SWV waveform is illustrated in Fig. 5a. As shown in Fig. 5b, molecularly dispersed CoPc on CNTs experiences a strong local electric field, resulting in a pronounced and well-defined Co(II)/Co(I) redox peak at around 0 V vs RHE, consistent with previous reports.³⁴ In contrast, despite identical CoPc loadings, the redox peak of the aggregated CoPc is significantly attenuated compared to that of the molecularly dispersed CoPc, indicating a suppression of the Co(II)/Co(I) redox response due to aggregation. When a potential is applied to the electrode, the electric field does not fully penetrate the aggregates, limiting effective potential transmission to the Co active sites and resulting in a substantially attenuated redox signal.

To further examine whether attenuation of the electric field shifts the rate-limiting step to CO₂ adsorption, we intentionally reduced the local electric field around molecularly dispersed CoPc by chelating Na⁺ in the electrolyte using the crown ether 18-crown-6.⁴⁹ Chelation of Na⁺ effectively increases the apparent cation size and expands the thickness of the outer Helmholtz layer,^{50,51} thereby reducing the potential gradient and interfacial electric field, as schematically illustrated in Fig. 5c. KIE studies were then performed on the CoPc/CNTs electrode in 1 M NaHCO₃/H₂O + 18-crown-6 and in 1 M NaDCO₃/D₂O + 18-crown-6 electrolytes. As shown in Fig. 5d, the partial current densities for CO production greatly decreased in the NaHCO₃ + 18-crown-6 electrolyte compared to those in NaHCO₃ (Fig. 3d), indicating reduced activity due to the weakened electric

field. Importantly, the CO₂RR activities in crown-ether-containing H₂O and D₂O electrolytes were nearly identical, giving KIE values of ~1 (Fig. 5e). These results indicate that upon weakening the electric field, the rate-limiting step shifts to CO₂ adsorption even for molecularly dispersed CoPc. In contrast, KIE values of ~3 were observed for the concurrent HER (Supplementary Fig. 14), confirming the reliability of KIE analysis in the presence of crown ether. Overall, these findings demonstrate that the Co active sites in aggregated CoPc experience a weakened electric field so that CO₂ adsorption becomes rate-limiting. This provides a clear mechanistic explanation for the dependence of the CO₂RR rate-limiting step on the dispersion state of CoPc.”

3. The "proton donor" function of anions near the electrode interface may be constrained by electric double-layer structure (e.g., specific adsorption competition). A Gouy-Chapman-Stern model analysis is needed to distinguish intrinsic proton-donating capacity from interfacial delivery efficiency.

Response: Yes, the intrinsic proton-donating capacity of the anions may not be very efficient and may be limited by their interfacial delivery efficiency, as they predominantly reside in the diffuse layer due to the electric double-layer structure. In contrast, cations with proton-donating ability can enter and accumulate within the electric double layer, allowing direct interactions with surface-bound intermediates and thereby enabling more efficient proton delivery. We have incorporated the following discussion into the revised manuscript:

*“Moreover, given the negatively charged cathode during CO₂RR, we expect electrolyte cations capable of donating protons to be more effective in promoting the rate-limiting protonation step. Such cations can enter and accumulate in the electric double layer, allowing direct interactions with surface-bound intermediates and thereby enabling more efficient proton delivery to facilitate *CO₂ protonation.”*

4. The HCO₃⁻ proton donor mechanism depends not only on pK_a but also on local pH, buffer capacity, and applied potential. In situ techniques (such as SEIRAS, pH-sensitive probes) should be employed to decouple these intertwined factors.

Response: We thank the reviewer for this comment. We agree that decoupling the proton-donation role of HCO₃⁻ from its pH-buffering behavior is essential for elucidating the origin of its promotional effect on CO₂RR with CoPc/CNTs. First, a comparison of the anion effects at various potentials, especially between HCO₃⁻ and H₂PO₄⁻/HPO₄²⁻, was included in Figure 6 and discussed in the manuscript. To evaluate the local pH and buffering capability of HCO₃⁻, we employed the *in situ* SECM equipped with a pH-sensitive probe to monitor and compare the local pH near the CoPc/CNTs electrode at various CO₂RR current densities in CO₂-saturated 1 M NaHCO₃ and 1 M NaClO₄ electrolytes, as shown in newly added Supplementary Figs. 12, 13, and the figure below. The local pH during CO₂RR in NaClO₄ increased substantially, from an initial bulk pH of 4.3 to a local pH of 8.6 at a current density of 5.0 mA cm⁻², reflecting the absence of buffering capacity in NaClO₄. In contrast, the pH change in NaHCO₃ was much smaller due to its buffering nature.

Then, to decouple the proton-donation effect of HCO₃⁻ from its pH-buffering capacity, we further compared the CO₂RR activities in NaHCO₃ and NaClO₄ using RHE-referenced potentials corrected by the measured local pH values, thereby removing pH-dependent thermodynamic contributions. As shown in the figure below (b), even after this correction, the CO₂RR activity in NaHCO₃ remained consistently higher than in NaClO₄, indicating that differences in pH-buffering

do not explain the variation in CO₂RR activity among the anions. Instead, the higher CO₂RR activity in NaHCO₃ is mainly attributed to the intrinsic proton-donation effect of HCO₃⁻, which enhances the *CO₂ protonation step on CoPc/CNTs.

The above figure has been added as Supplementary Figure 16 and the following discussion has been added to the revised manuscript: “*The proton-donating ability of the anions may not only facilitate protonation steps in CO₂RR, but also provide buffering capacity to mitigate local pH changes during reactions.^{61,62} To assess the contribution of such mechanistic factors, we employed the SECM with pH-sensitive probes to monitor the local pH during CO₂RR in different electrolytes. As shown in Supplementary Fig. 16a, the local pH near the CoPc/CNTs electrode during CO₂RR in NaClO₄ increased markedly, from an initial bulk pH of 4.3 to a local pH of 8.6 at a current density of 5.0 mA cm⁻², reflecting the absence of buffering capacity in NaClO₄. In contrast, the pH rise in NaHCO₃ was much smaller (Supplementary Fig. 13) due to its buffering nature. To account for pH-dependent thermodynamic effects when comparing CO₂RR activities, we re-calculated the activities using RHE-referenced potentials corrected by the measured local pH values (Supplementary Fig. 16b). Even after this correction, the CO₂RR activities in NaHCO₃ remained consistently higher than those in NaClO₄, indicating that differences in buffering or local pH do not explain the variation in CO₂RR activity among the anions. Instead, the higher CO₂RR activity in NaHCO₃ is mainly attributed to enhanced *CO₂ protonation on CoPc/CNTs due to the proton-donating ability of the HCO₃⁻ anion.*”

5. The difference of CO₂ solubility in D₂O versus H₂O may artificially reduce current density in isotope experiments. Normalizing j_{CO} values is essential for reliable KIE conclusions.

Response: We thank the reviewer for this comment. We agree that excluding the influence of CO₂ solubility or availability is essential for a reliable interpretation of the KIE results and the associated rate-limiting step of CO₂RR. The solubility of CO₂ at 25 °C and 1 atm is ~0.0337 mol L⁻¹ in both H₂O and D₂O (*J. Am. Chem. Soc.* **1938**, *60*, 2771–2773). Some literature reports CO₂ solubility on a mol kg⁻¹ basis, which introduces a slight difference between H₂O and D₂O, but this difference arises solely from the higher density of D₂O. When expressed on a mol L⁻¹ basis, which is the relevant metric for mass-transport considerations, the CO₂ solubility is essentially identical in the two solvents. Furthermore, all KIE measurements in our study were performed using 1 M bicarbonate electrolytes, where the equilibrium between dissolved CO₂ and bicarbonate species (CO₂ + H₂O ⇌ H⁺ + HCO₃⁻, CO₂ + D₂O ⇌ D⁺ + DCO₃⁻) continuously buffers and replenishes

CO₂ near the cathode. This equilibrium ensures a sustained and comparable CO₂ supply during CO₂RR in both H₂O and D₂O electrolytes. Therefore, the CO₂ solubility or availability can be excluded as the origin of the observed activity differences in the KIE studies.

We have added the following discussion to the revised manuscript: “*Prior to investigating the underlying mechanism, it is essential to establish that the observed differences in CO₂RR activity between NaHCO₃/H₂O and NaDCO₃/D₂O do not arise from extrinsic factors, such as differences in CO₂ solubility or the pK_a values of H₂O versus D₂O. First, the molar solubilities of CO₂ in H₂O and D₂O at 25 °C and 1 atm are both ~33.7 mmol L⁻¹,⁴¹ which cannot account for the pronounced CO₂RR activity differences. Moreover, in 1 M NaHCO₃/H₂O and 1 M NaDCO₃/D₂O electrolytes, the equilibrium between dissolved CO₂ and bicarbonate species (CO₂ + H₂O ⇌ H⁺ + HCO₃⁻, CO₂ + D₂O ⇌ D⁺ + DCO₃⁻) can continuously buffer and replenish CO₂ near the cathode, ensuring a sustained and comparable CO₂ supply for the CO₂RR. Consequently, CO₂ solubility or availability can be excluded as the origin of the observed activity differences in the KIE studies.*”

6. Anion comparisons in H-cells may not reflect practical electrolyzer conditions (flow cells, high current densities). Additional experiments are recommended to strengthen the applicability of the findings.

Response: We thank the reviewer for this comment. Following the suggestion, we tested CO₂RR on CoPc/CNTs in the presence of different anions using a flow cell equipped with gas-diffusion electrodes (GDEs), where CO₂ mass transport is greatly enhanced and high current densities are accessible. The CO₂RR performances, referenced to RHE, are shown in the figure below.

The CO₂RR activity follows the order: HCO₃⁻ > H₂PO₄⁻/HPO₄²⁻ > ClO₄⁻, consistent with the trend observed in H-cell, supporting the applicability of this anion-based strategy at high current densities. The consistently higher activity in HCO₃⁻ relative to ClO₄⁻ arises from its proton-donation role in the rate-limiting protonation step of CO₂RR, as well as a CO₂-shuttle role that sustains a higher local CO₂ concentration. The reduced performance gap between H₂PO₄⁻/HPO₄²⁻ and ClO₄⁻ at higher overpotentials likely arises from the stronger electrostatic repulsion of H₂PO₄⁻/HPO₄²⁻ by the negatively charged electrode, which diminishes its proton-donation contribution.

The above figure has been added as Supplementary Figure 17, with the following discussion added to the manuscript: *“These findings suggest a strategy for enhancing CO₂ electrolysis systems with a rate-limiting protonation step through the use of proton-donating anions.^{61,62} To further demonstrate this anion strategy and its applicability, we evaluated CO₂ electrolysis in presence of different anions using a flow cell equipped with gas-diffusion electrodes (GDEs), where the CO₂RR current density can be drastically increased due to enhanced mass transport.⁶³ Indeed, as shown in Supplementary Fig. 17, the NaHCO₃ electrolyte achieved the highest activity for CO₂RR on CoPc/CNTs, followed by NaH₂PO₄/Na₂HPO₄ and NaClO₄.”*

7. The focus on Co/Ni phthalocyanines lacks perspective on generalizability to other metal centers (Fe, Mn) or ligand frameworks (porphyrins, terpyridines). Expanding the discussion on design principles for broader applicability would enhance impact.

Response: We thank the reviewer for this comment. We have added the following discussion in the revised manuscript highlighting the broader applicability of our design principles:

“The above discussions focus on enhancing CO₂RR on CoPc with a rate-limiting protonation step by leveraging the proton-donating ability of electrolyte ions. More broadly, the methodology demonstrated here, which identifies whether the CO₂RR rate-limiting step involves proton transfer through KIE analysis and then optimizes the electrolyte accordingly, is applicable to other metal phthalocyanines with different metal centers (Ni, Fe, Cu, Mn) and to related molecular catalysts such as porphyrins and quaterpyridines. For catalysts with a protonation-limited pathway, CO₂RR performance can be enhanced by selecting electrolyte ions with appropriate proton-donating ability. Conversely, for systems where CO₂ adsorption is rate-limiting, CO₂RR activity can be improved by strengthening the interfacial electric field, for example through the use of partially or weakly solvated cations.⁵⁴ Together, these insights provide generalizable guidance for the rational design of efficient CO₂ electrolysis systems.”

8. All potentials were reported against the SHE. It is intriguing to know how the experiment setup was constructed and how different pH impact was introduced into the performance reporting.

Response: We thank the reviewer for this comment. In our study, all potentials were recorded using a leak-free Ag/AgCl reference electrode with *iR*-compensation applied using the dynamic Current-Interrupt method. The recorded potentials were subsequently converted to the SHE or the RHE scales. We agree with the reviewer that reporting CO₂RR activity on the RHE scale is more appropriate when comparing electrolytes with different pH values, because it removes pH-induced thermodynamic differences. Accordingly, we have reported the CO₂RR performance using RHE-referenced potentials in the revised manuscript.

First, for the KIE analysis in $\text{NaHCO}_3/\text{H}_2\text{O}$ and $\text{NaDCO}_3/\text{D}_2\text{O}$, we repeated CO_2RR tests on each catalyst (Au/C , CoPc/CNTs , NiPc/CNTs , the aggregated CoPc , and the CoPc/CNTs mixture) in CO_2 -saturated 1 M $\text{NaDCO}_3/\text{D}_2\text{O}$ while applying a negative potential shift of 24 mV relative to the $\text{NaHCO}_3/\text{H}_2\text{O}$ condition. This correction corresponds to $-0.0591 \times \Delta\text{pH} (\text{D}_2\text{O} - \text{H}_2\text{O})$, where $\Delta\text{pH} = 0.4$ based on the bulk pH values of 7.4 and 7.8 for CO_2 -saturated 1 M $\text{NaHCO}_3/\text{H}_2\text{O}$ and $\text{NaDCO}_3/\text{D}_2\text{O}$, respectively. We found the previously observed differences in CO_2RR activity between $\text{NaHCO}_3/\text{H}_2\text{O}$ and $\text{NaDCO}_3/\text{D}_2\text{O}$ for molecularly dispersed CoPc/CNTs and NiPc/CNTs remain after applying the pH-corrected potential shift, although the magnitude of the KIE values is slightly reduced. This persistence confirms a rate-limiting $^*\text{CO}_2$ protonation step on CoPc/CNTs and NiPc/CNTs . Furthermore, considering that local pH can shift from bulk pH during CO_2RR , possibly introducing additional thermodynamic shifts, we further used SECM equipped with pH-sensitive probes to monitor the local pH near the CoPc/CNTs electrode under *operando* conditions. As a result, while a slight increase in local pH was observed during CO_2RR in both $\text{NaHCO}_3/\text{H}_2\text{O}$ and $\text{NaDCO}_3/\text{D}_2\text{O}$ electrolytes, the relative difference between them remained ~ 0.4 pH unit, which is consistent with the initial bulk pH difference (7.4 versus 7.8). Because this initial difference was already accounted for in the RHE-referenced potentials, the minor local pH shifts do not exert any additional influence on the KIE analysis.

Second, for the study of electrolyte anion effects, we re-evaluated the CO_2RR performance of CoPc/CNTs in electrolytes with different anions under RHE-referenced conditions, as shown in the figures below.

At moderately negative potentials, where CO₂ depletion is not pronounced, the total current density on CoPc/CNTs, including both CO₂RR and HER, follows the trend: ClO₄⁻ < HCO₃⁻ < H₂PO₄⁻/HPO₄²⁻, consistent with the relative proton-donation capabilities indicated by their pK_a values. This supports the role of HCO₃⁻ and H₂PO₄⁻/HPO₄²⁻ as proton donors in the rate-limiting protonation step. To further distinguish the proton-donation role from the pH-buffering behavior, we monitored and compared the local pH near the CoPc/CNTs electrode at various CO₂RR current densities in CO₂-saturated 1 M NaHCO₃ and in 1 M NaClO₄ electrolytes using the SECM equipped with pH-sensitive probes. The local pH during CO₂RR in NaClO₄ increased substantially, from an initial bulk pH of 4.3 to a local pH of 8.6 at a current density of 5.0 mA cm⁻², reflecting the absence of buffering capacity in NaClO₄. In contrast, the pH change in NaHCO₃ was much smaller due to its buffering nature. Then, to decouple the proton-donation effect of HCO₃⁻ from its pH-buffering ability, we compared CO₂RR activities in NaHCO₃ and NaClO₄ using RHE-referenced potentials corrected by the measured local pH values, thereby removing pH-dependent thermodynamic contributions. Even after this correction, the CO₂RR activity in NaHCO₃ remained consistently higher than in NaClO₄, indicating that differences in pH-buffering do not explain the variation in CO₂RR activity among the anions. Instead, the higher CO₂RR activity in NaHCO₃ is primarily attributed to the intrinsic proton-donation effect of HCO₃⁻, which enhances the *CO₂ protonation step on CoPc/CNTs.

The updated and newly added figures, along with the following discussion, have been incorporated into the revised manuscript:

“Second, H₂O and D₂O exhibit a modest difference in their acid–base properties ($pK_w = 14.00$ for H₂O and 14.86 for D₂O), resulting in bulk pH/pD values of 7.4 and 7.8 for CO₂-saturated 1 M NaHCO₃ and NaDCO₃ electrolytes, respectively. In the above measurements, all potentials were converted to the RHE scale using the corresponding bulk pH, thereby accounting for pH-dependent thermodynamic effects in the comparison of CO₂RR activities. Nevertheless, local pH may change during reactions and influence reaction kinetics. To isolate genuine isotope effects from possible pH-induced artifacts, we employed in situ scanning electrochemical microscopy (SECM) to probe local pH under operando conditions. As shown in Supplementary Fig. 12, a customized SECM setup,⁴² equipped with an Au nanoelectrode tip whose exposed apex was functionalized with 4-hydroxylaminothiophenol (4-HATP)/4-nitrosothiophenol (4-NSTP), was used to measure the local pH near the cathode.^{43,44} A correlation between the mid-peak potential of the 4-HATP/4-NSTP anodic peak and the electrolyte pH was established as a calibration curve (Supplementary Fig. 12). Using this approach, we quantified the local pH near the CoPc/CNTs electrode under various CO₂RR current densities, as presented in Supplementary Fig. 13. While a slight increase in local pH was observed during CO₂RR in both electrolytes, the relative difference between them remained ~0.4 pH units, consistent with the initial bulk pH difference (7.4 versus 7.8). Because this initial difference was already accounted for in the RHE-referenced potentials, the minor local pH shifts do not exert any additional influence on the KIE analysis.”

“The proton-donating ability of the anions may not only facilitate protonation steps in CO₂RR, but also provide buffering capacity to mitigate local pH changes during reactions.^{61,62} To assess the contribution of such mechanistic factors, we employed the SECM with pH-sensitive probes to monitor the local pH during CO₂RR in different electrolytes. As shown in Supplementary Fig. 16a, the local pH near the CoPc/CNTs electrode during CO₂RR in NaClO₄ increased markedly, from an initial bulk pH of 4.3 to a local pH of 8.6 at a current density of 5.0 mA cm⁻², reflecting the absence of buffering capacity in NaClO₄. In contrast, the pH rise in NaHCO₃ was much smaller (Supplementary Fig. 13) due to its buffering nature. To account for pH-dependent thermodynamic effects when comparing CO₂RR activities, we re-calculated the activities using RHE-referenced potentials corrected by the measured local pH values (Supplementary Fig. 16b). Even after this correction, the CO₂RR activities in NaHCO₃ remained consistently higher than those in NaClO₄, indicating that differences in buffering or local pH do not explain the variation in CO₂RR activity among the anions. Instead, the higher CO₂RR activity in NaHCO₃ is mainly attributed to enhanced *CO₂ protonation on CoPc/CNTs due to the proton-donating ability of the HCO₃⁻ anion.”

9. Why was no liquid product detected from CoPc system?

Response: We thank the reviewer for this comment. To address the concern regarding CH₃OH production on CoPc/CNTs, we further performed bulk electrolysis of CO₂RR in 1 M NaHCO₃ at more negative potentials, as shown in the figure below. CoPc/CNTs indeed produces CH₃OH, but only within a narrow and more negative potential window (<-0.80 V vs RHE) and substantially lower Faradaic efficiencies compared to CO. Thus, under our experimental conditions, CH₃OH is not a major CO₂RR product on CoPc/CNTs. Interestingly, upon surveying the literature, we found pronounced variability in the reported CH₃OH production performance on CoPc. Even when CoPc is supported on CNTs, the reported partial current densities for CH₃OH vary widely (from negligible to ~10 mA cm⁻²), and the corresponding Faradaic efficiencies span from negligible to ~40% (*Nature* **2019**, 575, 639–642; *Nat. Catal.* **2024**, 7, 1000–1009; *ACS Catal.* **2024**, 14, 366–372; *ACS Appl. Energy Mater.* **2024**, 7, 3091–3098; *J. Mater. Chem. A* **2024**, 12, 31547–

31556). Such discrepancies suggest that CH₃OH formation on CoPc is highly sensitive to subtle variation in experimental conditions. While CH₃OH production on CoPc is an intriguing topic, a detailed investigation of this pathway is beyond the scope of our present study, which focuses on CO₂RR to CO on metal phthalocyanines and aims to elucidate the fundamental rate-limiting step.

We have added the above figure as Supplementary Fig. 7 and the following statement into the revised manuscript: “*In addition, methanol can be formed as a liquid product during CO₂RR on CoPc/CNTs,³¹ which was detected in 1 M NaHCO₃ electrolyte at more negative potentials (< -0.80 V versus RHE), as shown in Supplementary Fig. 7. However, methanol formation occurs only at high overpotentials with relatively low selectivity compared to CO production. Given that the present work is centered on CO₂RR to CO on metal phthalocyanines and aims to elucidate the fundamental rate-limiting steps governing this primary reaction pathway, the following studies will focus on the CO₂-to-CO conversion.*”

10. Were the CoPc and NiPc molecules molecularly dispersed or forming aggregates on the CNT surface? Detailed characterization results should be supplied.

Response: We thank the reviewer for this comment. Characterization of the dispersion state of CoPc and NiPc in the CoPc/CNTs and NiPc/CNTs samples was conducted, and the corresponding discussion is included in the “Characterization of CoPc/CNTs and NiPc/CNTs Catalysts” section of the original manuscript. For clarity and verification, we provide a brief summary of these results below. The molecular dispersion of CoPc and NiPc in the CoPc/CNTs and NiPc/CNTs samples is further supported by: (i) comparison of morphology with intentionally aggregated samples, and (ii) examination of their crystalline structures using XRD.

First, CoPc aggregates in the aggregated CoPc sample (prepared by drop-casting a DMF solution of CoPc onto an AvCarb GDS2230 substrate) and in the CoPc/CNTs mixture sample (prepared by grinding CoPc powders with CNTs) exhibits a nanoparticle or nanoflake morphology, with feature sizes ranging from sub-micrometers to several micrometers, as shown in the figures below.

The aggregated CoPc sample (Supplementary Fig. 8):

The CoPc/CNTs mixture sample (Supplementary Fig. 10):

In contrast, no such aggregated domains were observed in the CoPc/CNTs or NiPc/CNTs samples. SEM imaging over large areas, as well as TEM imaging combined with EDS mapping, consistently reveal a uniform dispersion on CNTs without any detectable CoPc or NiPc aggregates (see the figures below).

The CoPc/CNTs sample (Supplementary Fig. 1):

The NiPc/CNTs sample (Supplementary Fig. 2):

The CoPc/CNTs and NiPc/CNTs sample (Fig. 2):

Second, our XRD analysis confirmed the absence of molecular aggregation in the CoPc/CNTs sample. No diffraction peaks corresponding to bulk crystalline CoPc were observed, whereas a pronounced CoPc peak appears in the CoPc/CNTs mixture sample, as shown in the figures above. This structural evidence corroborates that CoPc is molecularly dispersed on the CNT support in the CoPc/CNTs sample.

Reviewer #3:

This manuscript examines the reaction mechanism, specifically the rate-limiting step, of the electrochemical conversion of CO₂ to CO using metal phthalocyanine catalysts. The rate-limiting step is identified as the protonation of adsorbed *CO₂ on dispersed CoPc-supported CNTs. In contrast, on aggregated CoPc, CO₂ adsorption becomes the rate-limiting step. The authors investigated how the dispersion rate and electrolyte components (anions and cations) affect the CO₂RR to CO on CoPc/CNTs, thereby guiding the rational design of molecular electrocatalysts. Overall, this manuscript offers sufficient evidence to clarify the rate-determining step of CO₂RR to CO on metal phthalocyanine catalysts. Major Revision. Some specific comments are provided below.

Response: We thank the reviewer for the positive evaluation of our work and we deeply appreciate the constructive comments and suggestions, which have helped us greatly improve the manuscript. In response, we have further performed new experiments with new data and discussions to deepen the mechanistic understanding and strengthen our manuscript. Our major revisions include:

- (1) We have gained a deeper understanding of the mechanism and attributed the shift in the rate-limiting step on aggregated CoPc to a weakened interfacial electric field at the Co active sites, as supported by solid evidence presented in Figure 5 of the revised manuscript.
- (2) We have employed *in situ* scanning electrochemical microscopy (SECM) to probe the local pH under *operando* conditions and have directly addressed all comments related to local pH, as demonstrated by the results in Supplementary Figs. 12, 13, and 16.
- (3) We have addressed other comments by incorporating new results and discussions in the revised manuscript.

Our point-to-point responses to the comments and corresponding revisions are described below. We believe that our manuscript has been greatly improved after the revisions. A copy of the revised manuscript with all changes highlighted using “Track Changes” is provided to facilitate review.

1. I would recommend that authors add in-situ microscopy evidence to monitor the reaction intermediates to support the mechanism study further.

Response: We thank the reviewer for this valuable comment. Following the reviewer’s suggestion, we performed additional mechanistic studies using *in situ* scanning electrochemical microscopy (SECM). In the revised manuscript, SECM is employed to monitor the local pH evolution near the cathode in H₂O and D₂O electrolytes, as well as in electrolytes containing different anions. These measurements allow us to directly probe and decouple the possible thermodynamic contributions arising from local pH shifts during CO₂RR. Consequently, the *operando* SECM results confirm a rate-limiting *CO₂ protonation step for CO₂RR on molecularly dispersed metal phthalocyanines, as well as the proton-donation role of HCO₃⁻ in facilitating this step. These findings substantially strengthen the mechanistic understanding in the revised manuscript.

First, isolating intrinsic isotope effects from pH-induced artifacts is essential for a reliable interpretation of the KIE results and the associated rate-limiting step of CO₂RR. We used SECM to probe the local pH near the CoPc/CNTs electrode under reaction conditions in the NaHCO₃/H₂O and NaDCO₃/D₂O electrolytes, as shown in the figure below. A customized SECM setup that was developed by our own group (*Ph.D. Dissertation*, 2023, University of Central Florida, Orlando, FL, <https://stars.library.ucf.edu/etd2020/1631/>) with an Au nanoelectrode tip functionalized with

4-hydroxyaminothiophenol (4-HATP)/4-nitrosothiophenol (4-NSTP) was used to measure the local pH near the cathode (*Anal. Chem.* **2020**, *92*, 2237–2243; *JACS Au* **2021**, *1*, 1915–1924). A correlation between the mid-peak potential of the 4-HATP/4-NSTP anodic peak and the electrolyte pH was established as a calibration curve (see the figure below).

Using this approach, we successfully measured the local pH near the CoPc/CNTs electrode under various CO₂RR current densities, as presented in the figure below. While a slight increase in local pH was observed during CO₂RR in both electrolytes, the relative difference between them remained ~0.4 pH units, consistent with the initial bulk pH difference (7.4 versus 7.8). Because this initial difference was already accounted for in the RHE-referenced potentials, the minor local pH shifts do not exert any additional influence on the KIE analysis. This indicates that the measured KIE primarily arises from an intrinsic isotope effect rather than from pH-induced thermodynamic artifacts, thereby confirming a rate-limiting *CO₂ protonation step on the molecularly dispersed metal phthalocyanines.

The above two figures has been added as Supplementary Figure 12 and Supplementary Figure 13, respectively. The following discussion has been incorporated into the revised manuscript:

“Second, H_2O and D_2O exhibit a modest difference in their acid–base properties ($pK_w = 14.00$ for H_2O and 14.86 for D_2O), resulting in bulk pH/pD values of 7.4 and 7.8 for CO_2 -saturated 1 M NaHCO_3 and $NaDCO_3$ electrolytes, respectively. In the above measurements, all potentials were converted to the RHE scale using the corresponding bulk pH, thereby accounting for pH-dependent thermodynamic effects in the comparison of CO_2RR activities. Nevertheless, local pH may change during reactions and influence reaction kinetics. To isolate genuine isotope effects from possible pH-induced artifacts, we employed *in situ* scanning electrochemical microscopy (SECM) to probe local pH under operando conditions. As shown in Supplementary Fig. 12, a customized SECM setup,⁴² equipped with an Au nanoelectrode tip whose exposed apex was functionalized with 4-hydroxylaminothiophenol (4-HATP)/4-nitrosothiophenol (4-NSTP), was used to measure the local pH near the cathode.^{43,44} A correlation between the mid-peak potential of the 4-HATP/4-NSTP anodic peak and the electrolyte pH was established as a calibration curve (Supplementary Fig. 12). Using this approach, we quantified the local pH near the CoPc/CNTs electrode under various CO_2RR current densities, as presented in Supplementary Fig. 13. While a slight increase in local pH was observed during CO_2RR in both electrolytes, the relative difference between them remained ~ 0.4 pH units, consistent with the initial bulk pH difference (7.4 versus 7.8). Because this initial difference was already accounted for in the RHE-referenced potentials, the minor local pH shifts do not exert any additional influence on the KIE analysis.”

Furthermore, decoupling the proton-donation role of HCO_3^- from its pH-buffering behavior is essential for elucidating the origin of its promotional effect on CO_2RR with CoPc/CNTs. To evaluate the buffering capability of HCO_3^- , we monitored and compared the local pH near the CoPc/CNTs electrode at various CO_2RR current densities in CO_2 -saturated 1 M NaHCO_3 and 1 M NaClO_4 electrolytes using the SECM equipped with pH-sensitive probes, with the results exhibited in the figure below.

The local pH during CO_2RR in $NaClO_4$ increased substantially, from an initial bulk pH of 4.3 to 8.6 at a current density of 5.0 mA cm^{-2} , reflecting the absence of buffering capacity in $NaClO_4$. In contrast, the pH change in $NaHCO_3$ was much smaller due to its buffering nature. Then, to decouple the proton-donation effect of HCO_3^- from its pH-buffering capacity, we compared the CO_2RR activities in $NaHCO_3$ and $NaClO_4$ using RHE-referenced potentials corrected by the measured local pH values, thereby removing the pH-dependent thermodynamic contributions. As shown in the figure above, even after this correction, the CO_2RR activity in $NaHCO_3$ remained

consistently higher than in NaClO₄, indicating that differences in pH-buffering do not explain the variation in CO₂RR activity among the anions. Instead, the higher CO₂RR activity in NaHCO₃ is primarily attributed to the intrinsic proton-donation effect of HCO₃⁻, which enhances the *CO₂ protonation step on CoPc/CNTs.

The above figure has been added as Supplementary Figure 16, and the following discussion has been added to the revised manuscript: “*The proton-donating ability of the anions may not only facilitate protonation steps in CO₂RR, but also provide buffering capacity to mitigate local pH changes during reactions.^{61,62} To assess the contribution of such mechanistic factors, we employed the SECM with pH-sensitive probes to monitor the local pH during CO₂RR in different electrolytes. As shown in Supplementary Fig. 16a, the local pH near the CoPc/CNTs electrode during CO₂RR in NaClO₄ increased markedly, from an initial bulk pH of 4.3 to a local pH of 8.6 at a current density of 5.0 mA cm⁻², reflecting the absence of buffering capacity in NaClO₄. In contrast, the pH rise in NaHCO₃ was much smaller (Supplementary Fig. 13) due to its buffering nature. To account for pH-dependent thermodynamic effects when comparing CO₂RR activities, we re-calculated the activities using RHE-referenced potentials corrected by the measured local pH values (Supplementary Fig. 16b). Even after this correction, the CO₂RR activities in NaHCO₃ remained consistently higher than those in NaClO₄, indicating that differences in buffering or local pH do not explain the variation in CO₂RR activity among the anions. Instead, the higher CO₂RR activity in NaHCO₃ is mainly attributed to enhanced *CO₂ protonation on CoPc/CNTs due to the proton-donating ability of the HCO₃⁻ anion.*”

2. Except for the effects of anions and cations, will the pH variation affect the rate-limiting step on metal phthalocyanine?

Response: We thank the reviewer for this comment. To evaluate whether electrolyte pH influences the rate-limiting step of CO₂RR on metal phthalocyanines, we performed additional KIE analysis on CoPc/CNTs in acidic media. We note that alkaline electrolytes are unsuitable for this analysis as the rapid reaction of continuously supplied CO₂ with OH⁻ steadily neutralizes the solution.

Specifically, CO₂RR was conducted on CoPc/CNTs in CO₂-saturated 1 M NaClO₄/H₂O and 1 M NaClO₄/D₂O electrolytes, both adjusted to pH/pD = 2. Note that bicarbonate solutions cannot be adjusted to such acidic conditions because of their buffering capacity. As shown in the figure below, the partial current densities for CO production in NaClO₄/H₂O were consistently higher than those in NaClO₄/D₂O, yielding KIE values of 1.3~1.7. This indicates that the rate-limiting step remains the *CO₂ protonation step on CoPc/CNTs in acidic media, and it is not altered by changing the electrolyte pH from near-neutral to acidic conditions.

We have added the above figure as Supplementary Figure 6 and the following statement into the revised manuscript:

“The above studies were performed in near-neutral electrolytes, while electrolyte pH may also influence the rate-limiting step of CO₂RR on metal phthalocyanines. Therefore, we further examined the rate-limiting step of CO₂RR on CoPc/CNTs in acidic media (1 M NaClO₄/H₂O and 1 M NaClO₄/D₂O, pH/pD = 2) using KIE analysis. As exhibited in Supplementary Fig. 6, the partial current densities for CO production in NaClO₄/H₂O were consistently higher than those in NaClO₄/D₂O, yielding KIE values of 1.3~1.7. This indicates that the rate-limiting step of CO₂RR on metal phthalocyanines is not altered by pH variation.”

Reviewer #4:

Response: We thank the reviewer for co-reviewing our manuscript and for offering constructive comments that have strengthened the quality of our work.

Response to Referees

Reviewer #1:

The authors have improved the quality of their manuscript by conducting new experiments and clarifying further mechanistic understanding. The local pH measurement by SECM is highly recommended. I am fine with most of their responses. Significant improvement can be seen in the revised manuscript compared to the original one.

Response: We sincerely thank the reviewer for the positive evaluation of the revised manuscript. We appreciate your recognition that our new experiments and discussions have strengthened the mechanistic understanding and improved the overall quality of the manuscript. We appreciate the time and effort you invested in the review process and your constructive comments, which have greatly improved the quality of our work.

Regarding to the product selectivity of CO versus methanol over CoPc with different supports, desperation and even Nafion additives, I would suggest the authors to provide more robust evidences, as this is an important unaddressed issue in the field. While the reported methanol FE in this work is low even at very negative potentials, higher methanol selectivity at less negative potentials is often reported in literature. Insights into such divergence would make much sense.

Response: We thank the reviewer for this comment and suggestion. We agree that there seems to be a divergence between our results and some prior reports regarding CO₂RR selectivity toward methanol (CH₃OH) on CoPc/CNTs. However, we would first like to emphasize that the potential window and the optimal potential (around -1.0 V vs RHE in near-neutral electrolytes) for CH₃OH production observed in our study are consistent with recent reports (*Nat. Catal.* **2024**, *7*, 987–999; *Angew. Chem. Int. Ed.* **2024**, *63*, e202310623; *J. Mater. Chem. A* **2024**, *12*, 31547–31556; *ACS Appl. Energy Mater.* **2024**, *7*, 3091–3098). Actually, CH₃OH formation at less negative potentials has been primarily reported when CO is used as the initial reactant (*Nat. Synth.* **2023**, *2*, 1194–1201; *ACS Catal.* **2024**, *14*, 366–372). This difference is likely associated with the relay pathway on CoPc (CO₂ → CO → CH₃OH), where direct CO electrolysis provides a higher CO availability and eliminates competition from CO₂ adsorption, thereby facilitating CO reduction to CH₃OH.

To explain and clarify the performance gap in CO₂RR to CH₃OH, we systematically optimized several experimental parameters, including the support for CoPc, the Nafion content in the catalyst ink, and the catalyst loading, as suggested by the reviewer. Regarding the dispersion of CoPc, prior studies show that aggregated CoPc greatly suppresses CH₃OH formation relative to molecularly dispersed CoPc, an effect attributed to weakened cation stabilization of key reaction intermediates (*Nat. Catal.* **2024**, *7*, 987–999). In our work, CoPc is already molecularly dispersed on CNTs, so we did not further modify the dispersion state of CoPc here.

We next compared CH₃OH production on CoPc supported on CNTs (CoPc/CNTs) and carbon black (CoPc/CB) at representative potentials. As included in our response to the first-round review, CoPc/CB exhibits slightly lower activity but comparable selectivity for CH₃OH production relative

to CoPc/CNTs, as shown in the figure below (a,b). Therefore, under our experimental conditions, the effect of the carbon support on CH₃OH production is not apparent.

We subsequently evaluated the influence of the Nafion content in the catalyst ink on CH₃OH production. As presented in the figure above (c,d), both the partial current density and Faradaic efficiency for CH₃OH remain largely unchanged as the Nafion amount increases from 2.5 to 15 $\mu\text{L mg}^{-1}_{\text{CoPc/CNTs}}$. The results suggest that, within the range examined, the Nafion content does not play a critical role in determining CH₃OH selectivity.

Finally, we examined the effect of catalyst loading on CH₃OH production, as presented in the figure above (e,f). Notably, increasing the total catalyst loading results in an apparent increase in CH₃OH production. This trend is consistent with prior reports (*Angew. Chem. Int. Ed.* **2024**, *63*, e202310623; *J. Mater. Chem. A* **2024**, *12*, 31547–31556) and can be rationalized by enhanced CO adsorption and subsequent reduction when competition from CO₂ is alleviated per Co active site. Specifically, as catalyst loading increases, the average CO₂ flux per CoPc site decreases, favoring CO adsorption on CoPc and its subsequent reduction to CH₃OH.

As a result of these optimizations, CO₂RR on CoPc/CNTs achieved a partial current density of ~8 mA cm⁻² and a Faradaic efficiency >20% for CH₃OH production under our experimental conditions. This performance is comparable to recent reports, such as 7–10 mA cm⁻² partial current density with ~30% Faradaic efficiency for CH₃OH production by Shao-Horn et al. (*Nat. Catal.* **2024**, *7*, 1000–1009), and ~8 mA cm⁻² partial current density with ~30% Faradaic efficiency for CH₃OH production by Wang et al. (*Nat. Nanotechnol.* **2025**, *20*, 515–522).

We have further strengthened the discussion of CH₃OH production in the revised manuscript and cited relevant references to support this discussion, as follows: “*In addition, methanol can be formed as a liquid product during CO₂RR on CoPc/CNTs,³¹ which was detected in 1 M NaHCO₃ electrolyte at more negative potentials (< -0.80 V versus RHE), as shown in Supplementary Fig. 7. However, methanol formation occurs only at high overpotentials with relatively low selectivity compared to CO production. The possible divergence between our methanol production performance and prior reports likely arises from the high sensitivity of this pathway to experimental parameters, such as CoPc-to-CNT mass ratio, catalyst loading, and CO₂ partial pressure.^{31,33–35} Given that the present work is centered on CO₂RR to CO on metal phthalocyanines and aims to elucidate the fundamental rate-limiting steps governing this primary reaction pathway, the following studies will focus on the CO₂-to-CO conversion.*”

We agree with the reviewer that “this is an important unaddressed issue in the field”. However, this present work is centered on CO₂RR to CO on metal phthalocyanines and aims to elucidate the fundamental rate-limiting steps governing this primary reaction pathway. Therefore, we prefer not to incorporate the additional figure and related CH₃OH results into the current manuscript, as doing so may expand the scope beyond the central mechanistic objectives. We appreciate the reviewer’s suggestion and will address CH₃OH production on CoPc in our future work.

Reviewer #2:

The authors have provided comprehensive and thorough responses to the raised questions, supplemented with robust evidence. The manuscript has been substantially improved and is now recommended for publication.

Response: We sincerely thank the reviewer for the positive evaluation of the revised manuscript. We appreciate your recognition of the quality of our responses and supporting evidence. We also appreciate the time and effort you invested in the review process and the constructive comments, which have greatly improved the quality and depth of our work.

Reviewer #3:

They have done an excellent work for the revision. I do not have extra comments this time. Please make a decision for this manuscript.

Response: We thank the reviewer for the positive evaluation. We appreciate the time and effort you invested in the review process and your constructive comments, which have helped improve the quality of the manuscript.

Reviewer #4:

Response: We thank the reviewer for co-reviewing the manuscript. We appreciate the time and effort you invested in the review process and your constructive comments, which have helped improve the quality of the manuscript.

Response to Referees

Reviewer #1:

The authors have addressed my comments. The current manuscript is now suitable for publication in Nature Communications.

Response: We thank the reviewer for the positive evaluation. We are grateful for your time and constructive input throughout the review process, which have greatly improved the quality of our work.